# Ultrasonic Guided-Waves Sensors and Integrated Structural Health Monitoring Systems for Impact Detection and Localization: A Review

**DOI:** 10.3390/s21092929

**Published:** 2021-04-22

**Authors:** Lorenzo Capineri, Andrea Bulletti

**Affiliations:** Department of Information Engineering, University of Florence, Via S. Marta 3, 50139 Firenze, Italy; andrea.bulletti@unifi.it

**Keywords:** structural health monitoring (SHM), acoustic emission, guided waves, Lamb waves, sensors, ultrasound, piezoelectric, composites, piezopolymers, PVDF, interdigital transducer (IDT), PWAS, CMUT, mems, analog electronic front end, analog signal processing, impact localization, impact detection, sensor node, wireless sensor networks (WSN), IoT, deep learning, artificial intelligence

## Abstract

This review article is focused on the analysis of the state of the art of sensors for guided ultrasonic waves for the detection and localization of impacts for structural health monitoring (SHM). The recent developments in sensor technologies are then reported and discussed through the many references in recent scientific literature. The physical phenomena that are related to impact event and the related main physical quantities are then introduced to discuss their importance in the development of the hardware and software components for SHM systems. An important aspect of the article is the description of the different ultrasonic sensor technologies that are currently present in the literature and what advantages and disadvantages they could bring in relation to the various phenomena investigated. In this context, the analysis of the front-end electronics is deepened, the type of data transmission both in terms of wired and wireless technology and of online and offline signal processing. The integration aspects of sensors for the creation of networks with autonomous nodes with the possibility of powering through energy harvesting devices and the embedded processing capacity is also studied. Finally, the emerging sector of processing techniques using deep learning and artificial intelligence concludes the review by indicating the potential for the detection and autonomous characterization of the impacts.

## 1. Introduction

Structural health monitoring (SHM) is a rapidly evolving field, and there is a vast literature covering several topics that are related to this field, including several excellent reviews. The motivations of this paper are to report the recent developments on technologies, especially sensors and mixed signal electronic interfaces, which enable integration into a sensor node. The sensor node concept is analyzed in this review and the perspective for integrating with monitored facilities is examined. In the introduction the main concepts behind the design of a SHM system for impact monitoring and main review papers are reported. Later in the introduction, the main system components are defined and, in the following sections, they will be discussed more deeply.

Ultrasonic non-destructive investigation (NDI) methods that are based on the principle of acoustic emission (AE) have evolved over the past two decades towards structural monitoring systems with guided ultrasonic waves [1,2], driven by applications in the aerospace, civil engineering, energy conversion, and transportation systems automotive (e.g., wind turbines, pipelines, and liquid natural gas cylinders). The safety of the structure and the prediction of in-service period are the key elements that must be provided by SHM and the underlying theory about these topics were explained in a comprehensive work by Farrar and Worden [3] and in a related book [4]. Structure damaging can occur for different causes (e.g., breakages due to fatigue, mechanical and thermal stresses, impacts with objects, etc.), and their consequences often are not optically visible or detectable. The damage is sometimes not visible, because it is internal to the structure or small but not without importance from the point of view of the safety and reliability of the operation of the system. To avoid catastrophic accidents, the damage prognosis is an essential task that is connected to the impact events; a framework for the damage prognosis was described in chapter 14 of the book that was published by Farrar and Worden [5].

Non-Destructive Testing (NDT) is a wide group of analysis techniques used in science and technology industry to evaluate the properties of a material, component, or system without causing damage, and it is often carried out in laboratory or on site on a scheduled program. SHM, unlike NDT, requires the installation of sensors/transducers operating in the environment in which the structure operates under remote control and for this reason the realization of such systems requires a considerable effort of integration of several disciplines:(1)modelling of damage physical phenomena and their influence on the physical sensed quantities,(2)sensors, including calibration and self-diagnostics,(3)front-end electronics including embedded processing,(4)data transmission (wired, wireless),(5)online (or real time) or offline signal/image processing,(6)impact event detection and localization(7)damage detection and classification techniques that are based on database processing,(8)prognostics,(9)artificial intelligence (AI)/machine learning (ML) for automatic damage detection and progression evaluation.

Figure 1 illustrates the different components of an SHM system and their interaction: the environmental conditions, the on-site hardware, and the off-site hardware and software resources. The different characteristics of the structures (dimensions, materials, and environmental conditions) and their structural monitoring systems (cost, footprint, weight, power consumption, safety and reliability criteria, and response/update times) often require the design of ad hoc systems by exploiting multidisciplinary knowledge in electronics, informatics, telecommunications, and, finally, material technology and mechanical properties.

For a general understanding of the state of the art, the reader can refer to the review paper of Mitra et al. [6], where several publications relating to the various components of an SHM system are discussed (see Figure 1); in the paper of Mitra et al. [6], various monitoring techniques based on ultrasonic guided waves (UGW) piezoelectric and fiber optic sensors, laser vibrometry (SLDV) techniques are examined. In addition, indications are given of what research and development lines may be for advanced SHM systems. As already introduced in this paragraph, monitoring techniques that are based on UGW by piezoelectric transducers are among the most common and most developed, since they have a longer history [7] than SHM systems based on optical sensors, in particular Fiber Bragg Grating (FBG) sensors; for completeness, the evolution of state of the art for optoelectronic sensors is reported in [8,9,10,11,12], but it is not discussed further in this paper. Similarly, the evolution of piezoelectric materials for the construction of UGW sensors and transducers, the development of low-power consumption integrated electronic components and systems requires a continuous updating of research to provide new design methodologies and technologies to bring in the field the SHM systems. Although many published papers report the outcomes that were obtained with laboratory set-up of guided ultrasonic wave SHM systems, their demonstration in the field is still limited. For the latter problem, there are various reasons, but certainly one of these is the complexity of the installation of the sensors on a target structure, the real time signal acquisition and processing, and the replication of the real-life environmental conditions. An interesting reference for the testing of SHM systems in the aerospace industry is provided in a report that was presented by Dennis Roach of Sandia National Labs [13]: this report shows the objectives and implementations of SHM systems for airplanes and includes several examples with piezoelectric and fiber optic sensor applications for monitoring impacts, deformations, debonding, delamination, and damage progression.

Finally, it is useful to point out the effort made to create standards for the development of systems and methods for SHM and NDT based on acoustic emission, especially for the rapidly evolving SHM sector; a comprehensive reference is the British Standard for Acoustic Emission and Condition Monitoring that was published in The Official Yearbook of the British Institute of Non-Destructive Testing [14]. Some main book references on SHM based on UGW can be found in [15,16,17,18,19]. After this introduction of the background of SHM systems that are based on UGW in active and passive modes, the present paper focusses the elements of the system that is shown in Figure 1 for the implementation of impact monitoring advanced systems on metal and composite materials with UGW piezoelectric sensors. In this paper, we primarily consider piezoelectric sensors used for impact detection in passive (“listening”) mode, but also in combination with the transducers operating in active mode for the investigation of damage and its progression over time. The trend of integrating different sensor types (UGW, FBG, accelerometer, strain, temperature, etc.) into a node increases the information regarding the impact and the operational conditions of the sensors that are influenced by the environment, leading to the concept of a “multifunctional sensor node”.

The evolution from the common AE monitoring configuration with a layout of sparse single element sensors with off-the shelf electronics to the recent design of sensors networks with “smart-sensor nodes” requires a continuous analysis and evaluation of the progresses in several fields.

This work first presents a review of methodological developments about the criteria to be adopted for the elaboration of impact-generated Lamb wave modes (Section 2). Subsequently, it addresses technological developments about UGW sensors and actuators including new materials and sensor types with a focus on microfabrication technologies (Section 3), front-end analog-digital electronics and power management (Section 4), and the integration wired or wireless sensor networks (WSN) with real-time acquisition and signal processing capabilities for monitoring environmental parameters (Section 5). Finally, the authors believe that it is important to report, in Section 6, the recent applications of Artificial Intelligence (AI) and Machine Learning (ML) techniques for the detection and autonomous positioning of impact events. In the Conclusions, we will draw guidance on research topics and challenges for the various areas that are covered by Section 2, Section 3, Section 4, Section 5 and Section 6. A list of acronyms is given at the end of the paper to facilitate the reader interested mainly in some of the topics that are addressed in this article; the list also shows the acronyms that have been recently introduced into the literature by the new technologies and methodologies that were adopted in this multidisciplinary field with which the reader can become familiar.

## 2. Characteristics of Signals Generated by Impacts on Planar Structures Relevant to the Design of SHM Systems

### 2.1. Dispersion and Attenuation of Lamb Waves

In this section, the implication of the attenuation and dispersion characteristics of UGW relevant for the design and implementation of a SHM system are discussed. The interested reader can find main references for the theory and modelling of ultrasonic guided waves [1,20]. In brief, we point out that ultrasonic waves that are guided for SHM are mechanical waves that propagate within a material delimited by an interface with a different medium. Propagation within the space-limited structure simultaneously produces dispersive modes of propagation in frequency. In the case of structures with thicknesses comparable to wavelength, such as thin planar structures, propagation modes have symmetrical and antisymmetric characteristics with respect to the axis of symmetry of the structure and they are determined by the theory behind Lamb waves, as explained in [21]. For an isotropic and homogeneous laminate material (e.g., aluminum), we illustrate the dispersion characteristics in Figure 2 (top) by the calculated phase velocities for the different guided modes versus the frequency x thickness product (fxd). Another difference between these two UGW modes is the dependence on frequency attenuation, as shown in Figure 2 (bottom): the S_0_ mode is remarkably attenuated in the low frequency range and for the reception of this mode is necessary a high pass filtering and amplifier gain to be separated from the slower and higher amplitude components of the A_0_ mode.

Therefore, the propagation of symmetrical modes within a planar structure is a two-dimensional phenomenon; the propagation of the various modes is subjected to attenuation that mainly follows the law of geometric decay inversely at the root of the distance. The authors in [20] proposed a deep and comprehensive analysis of the attenuation phenomena that are basic in differentiating the design of SHM systems according to the characteristics of the different materials (composite or metallic) and the size of the structure; thus, attenuation analysis is essential in defining the distance and area coverage with a certain type of transducer/sensor without exceeding the attenuation limit (50–70 dB), which results in being difficult to deal with analog-front-end (AFE) electronics based on COTS, unless it has an acceptably expensive and complex electronic customized design. Indicatively, the operating frequencies for Lamb’s guided ultrasonic waves range from 100 kHz to 1 MHz and, in this wide range, a compromise must be found between attenuation, wavelength, minimum detectable impact energy, and for the transducers/sensors, the size, type, sensitivity, and bandwidth. To solve these problems, methods for optimizing the position of transducers have recently been proposed by Mallardo et al. [22] based on the background of UGW propagation theory; in this work, a method is developed to define the optimal positions considering the characteristics of the material and sensors, thus also optimizing the number of sensors transducers, while concluding that there is no general solution to the problem, since each application has different constraints and, therefore, requires a series of a priori choices.

### 2.2. Ultrasonic Guided Waves Generated by Different Velocity of Impacts on Isotropic Elastic Plates

Impact monitoring systems can be designed for different applications, where impacts with different objects hitting the structure have different energy, mass, and velocity. It is of interest to explain the different effects on UGWs that were generated by impacts at different velocity. There are several categories of impact loading: low velocity (large mass), intermediate velocity, high/ballistic velocity (small mass), and hyper velocity impacts. These categories of impact loading are important because there are remarkable differences in energy transfer between the object and target, energy dissipation, and damage propagation mechanisms as the velocity of the object varies. Low velocity impacts occur typically at a velocity below 10 m/s, intermediate impacts occur between 10 m/s and 50 m/s, high velocity (ballistic) impacts have a range of velocity from 50 m/s to 1000 m/s, and hyper velocity impacts have the range of 2 km/s to 5 km/s, according to the literature [23].

In several studies [19,20,21], the signals generated by non-destructive impacts have been treated, which is impacts that do not cause any damage to the laminate under examination. These papers consider single and multiple impacts, but detailed information on the energy characteristics is not provided regarding the impacts analyzed. Furthermore, in [22], the impacts are distinguished based on the potential energy of the impacting bodies, with values ranging from 500 mJ to 3.5 mJ. In other early studies on this subject [23,24], the impacts are instead distinguished based on the impact velocity. In several studies [24,25,26], signals that are generated by non-destructive impacts have been treated, which is, they do not cause any damage to the laminate under examination, neither with single impact nor with multiple impacts, however no information is given on the extent of impact. Furthermore, in [27], the impacts are distinguished based on the potential energy of the impacting bodies, with values ranging from 500 mJ to 3.5 mJ. In other early studies on this subject [28,29], the impacts are instead distinguished based on the impact velocity.

The study of impacts that occur in an isotropic elastic flat plate is based on following assumptions:The ultrasonic signal that is generated by an impact is a guided wave signal that propagates into the plate without energy loss [24,30].The frequency content of the ultrasonic signals that are generated by impacts depends on the impact velocity [28,31] and it is not modified during the propagation inside the plate [32].

According to the above assumptions, we can remark that the main feature of the signals generated by impacts is the impact velocity that also determines the amplitude of the Lamb waves. From the physics laws for a falling body from a certain height, the potential energy is converted to kinetic energy; the impact velocity *v_i_* can be calculated by knowing the kinetic energy *E_k_* and the mass *m* of the impacting object, as reported in the following formula:(1)$$v_{i}=\sqrt{\frac{2E_{k}}{m}}$$

The study reported in [28] shows that two fundamental propagation modes can be distinguished in impact phenomena: a slow propagation mode (flexural mode or A_0_ mode) and a fast propagation mode (extensional mode or S_0_ mode).

The amplitude of the signal in A_0_ mode is dominant as compared to the S_0_ mode, but the amplitude of the latter is strongly linked to the speed of the impact: the greater the speed of the impact the greater the amplitude of the signal relative to the S_0_ mode. The authors in [28] also reported an acquired signal from a high-speed impact (700 m/s), where they demonstrate, when that applying a low-pass filter with (with a cut-off frequency of 800 kHz), it is possible to only extract the two fundamental propagation modes (A_0_ and S_0_) and, in this case, the amplitude of the S_0_ mode becomes comparable to that of the A_0_ mode. According to the authors experience, we investigated the possibility to also extract the S_0_ mode signal in low velocity impacts by applying a low-pass filter in the analogic front-end electronic board with proper cut-off frequency. Figure 3 shows ultrasonic signals that are generated by a low-velocity impact (about 3 m/s) on an aluminum plate with thickness 1.5 mm.

From the analysis of Figure 3, it is apparent that the fast propagation mode S_0_ becomes comparable in amplitude with the A_0_ mode only after filtering the ultrasonic propagating signal that is generated by the impact. The possibility of processing the fast S_0_ mode instead of the slower A_0_ mode, is often the best signal processing design strategy, because this early arrival time signal is less affected by overlapping of the multiple reflections from the structure edges [33]; moreover, the impact signal detection and positioning is even more complicated in large structures for the higher attenuation and the mode conversions after the propagation on areas with different thicknesses. The topics briefly reviewed in this section remark the importance of the understanding the physical background for designing sensors and the analog front-end to simplify and make the information extraction from the signal reliable.

### 2.3. Signal Processing Techniques for Dispersion and Environmental Factors Compensation

From the preliminary considerations in the Introduction, we can remark that the rapid evolution towards integrated-SHM (ISHM) systems operating in different environmental conditions follows a different path than the common AE and NDT techniques, which use volumetric longitudinal or transverse ultrasonic waves with piezoelectric transducers that are connected to portable instruments and the region of interest (ROI) manually scanned of by a trained operator [34]; main differences are found for the signal processing adopted for both passive and active mode operation of the SHM system. The analysis of information gathered by a sensors layout due to the interaction between the UGW dispersive modes and the various types of structures is certainly a challenging aspect from the point of view of signal processing techniques that are based on the Continuous Wavelet Transform (CWT) or the Short Time Fourier transform (STFT). CWT decomposes a time domain signal into components that correspond to a frequency band. Each of these components contains a further temporal discretization. The resolution of the temporal discretization varies with each frequency component, resulting in a multi-resolution temporal frequency analysis. Because the modes S_0_ and A_0_ propagate with different amplitudes in the useful band and with different propagation speeds (see Figure 2), the CWT allows for a representation capable of separating the two contributions in different instants of time. One of the limitations of the CWT is the compromise between resolution in frequency and in time and, moreover, the calculation algorithm requires considerable computational resources, not always available within a sensor node. Alternatively, the simplest form is represented by the STFT, but, differently from CWT, does not have the possibility to be implemented with the multi-resolution functionality in the time/frequency domain. For example, the separation of the two modes S_0_ and A_0_ by CWT o STFT is relevant for the evaluation of the DToAs for low and high velocity impacts, as we will describe in Section 2.2. However, simple analysis with CWT or STFT may still be too restrictive in the presence of structures with inserts, reinforcement elements, and therefore several methods have recently been proposed to overcome this problem, such as those reported in [30,31,32,35,36,37]. The well-known time-reversal approachis another important method introduced in [38] to compensate for the dispersion and alleviate the complexity of Lamb wave signal interpretation; this approach was adopted by Zeng et al. [39]. The dispersion of the generated modes by impacts influences the spatial resolution of the adopted localization algorithm, because the propagation on long distances (e.g., meters) [40] on the plate elongates the initial wavelet. The mitigation of this problem can be done using algorithms that can process the received signals by compensating the phase delay according to the theory of UGW [1]. There are available efficient algorithms for this task, such as multiple signal classification (MUSIC) and RAPID [41]. New developments that are based on the MUSIC algorithm have been proposed for impact energy estimation [42] and for the direction of arrival of a Lamb wave [43]. The computational efficiency is also important for real-time systems and Zhong et al. proposed an improvement in the processing scheme [44].

The UGWs used in active mode for damage assessment have a great sensitivity to detect internal damage into the structure, and this is one of the main reasons of successful application of this NDT technique. The detection is often implemented on a data driven approach, where the received UGWs from a sensor layout are compared with a baseline of data acquired with a pristine structure. This approach is also rather simple to implement in sensors with on board embedded processing, but it suffers from the sensitivity to environmental and operational conditions, mainly temperature variations. Recently, Mariani et al. [45,46] have proposed a method for the compensation of this detrimental phenomenon. For the electro-mechanical-impedance (EMI) method, the temperature compensation was achieved with some benefits by using artificial neural network (ANN) as reported by Sepehry et al. [47].

### 2.4. Advanced Methods for Impact Detection and Localization

In general, impacts on a thin planar structure generate guided waves modes that can propagate away from the impact point. The localization of the impact point is commonly achieved by adopting a triangulation algorithm with at least three passive ultrasonic sensors being deployed on the planar structure. The accuracy of the impact point estimation depends on the estimates of the guided modes velocity and the measured differential time of arrival (DToA) among the sensors [48]. Recently, several papers have been published to improve the reliability and accuracy of impacts on complex structures other than from the simple panels often used by researchers in laboratory for calibration and performance assessment of a SHM system. The Akaike Information Criterion (AIC) criterion for the accurate estimation of DToA has been demonstrated by De Simone et al. [49]. Further research work has consolidated the investigation of the advantages of AIC, and a modified version for impact monitoring has been recently proposed by Seno et al. [50]. In the latter work, an ANN was trained for automatic classification of defects in composite materials that were tested in laboratory and simulated operational conditions. As already reported in the Introduction, Ono reports an extensive review of AE physical parameters for SHM systems in [20]. The characteristic of UGW generated by impacts has been outlined in Section 2.1 and Section 2.2. Such guided wave modes propagating into the planar structure mix-up due to the phase velocity dispersion and, in addition, the reflection phenomenon from the edge or from inserts or stiffening material or defects [51]. Moreover, mode conversion can occur when the ultrasonic guided waves travel across a discontinuity of acoustic properties in the planar structure, for example, a change in thickness or material composition. In general, the wave shape of the impact generated UGW is complex, but a list of features supported by theoretical modelling developed by Hakoda et al. [52] based on the phase velocity analysis can be derived. It is worth noticing that the propagation velocity analysis, in general, is more complex for a composite three-dimensional structure than the simpler case shown in Figure 2; even the example of time domain signals generated on an aluminum plate reported in Figure 3 is a simplified scenario with respect to real-life cases. In the following, we report two main considerations that are starting guidelines for the impact signals processing:(1)the early part of the signal consists of the fast phase velocity modes, typically the S_0_ mode in the low frequency range below the cut off frequency × thickness product (e.g., equal to 1.5 MHz × mm in Figure 2).(2)in the later part of the signal the contribution comes from slower modes that show also dispersion effect as for the A_0_ mode [53] or signals that travelled along longer paths or multiple reflections.

We can observe that the S_0_ mode being faster than A_0_ it is less prone to being overlapped by delayed signals, but, due to the greater attenuation at low frequencies, the S_0_ mode has a lower amplitude than the A_0_ mode; the higher velocity of this mode also implies that the error on its DToA estimation causes higher spatial errors in the triangulation algorithms or any other positioning method based on DToA [54,55,56]. The theory of UGW in a plate like structure also considers other types of waves than Symmetrical and Antisymmetrical Lamb wave modes: the shear horizontal (SH) mode. This is a non-dispersive mode and piezoelectric sensors/transducers can be designed to convert this wave type into voltage signals. Ren and Lisseden [57] have demonstrated the capability of also sensing Lamb waves that are of interest for impact detection in passive mode. Altammar et al. [58] studied the actuation and reception of shear modes by exploiting the d_35_ piezoelectric coefficient of lead zirconate titanate (PZT) sensors that were embedded in a laminate structure. d_35_ PZT is a class of PZT piezoelectric transducers that, when polarized along their thickness, they induce shear strain in the piezoelectric material. It is interesting to observe that the shear deformation has a stronger coupling coefficient (d_35_) than the common d_33_ or d_31_, indicating that d_35_ PZTs have stronger electromechanical coupling for sensing and actuation.

In the final part of this section, we review the advancements on signal processing techniques for anisotropic plate-like material. Anisotropic characteristics of composite structure require the adaption of impact positioning algorithms that were developed for isotropic plate like materials. The early research on signal processing techniques for isotropic metallic plates and anisotropic composites can be found in [26,56,59,60]. More recently, the signal processing techniques have been progressed to account for the UGW dispersion (see Section 2.1) and anisotropy of different type of composites, like unidirectional, quasi-isotropic composite fiber reinforce polymer (CFRP), and honeycomb, which are of interest for aerospace industry [39,49,56,61,62]. An early work of Scholey and Wilcox in 2010 [63] addressed the problem of impact detection on 3D structures and, recently, Moron et al. in 2015 [64]. Lanza di Scalea et al. published a work [65] for impact monitoring in complex composite material structure with an algorithm that is based on the rosette sensor configuration; this model-based approach could solve the problem of variation of phase velocity along different direction of a composite material.

## 3. Sensors and Transducers for Impact Monitoring

Piezoelectric sensors are common devices for the passive detection of impacts on the structure [7]. However, an SHM system can also operate in active mode with piezoelectric transducers for generating UGW for damage evaluation because of impact events. In this way, there is interest for a dual use of the transducers for both passive and active operation with an advantage for the reduction of system complexity. In this section, we will revise the main characteristic of sensors and some considerations regarding how to use transducers in passive mode are reported.

### 3.1. Single Element Piezoelectric Sensors for Impact Detection and Emerging/New Sensing Materials

The piezoelectric sensors commonly used for reproducing the impact stress waves in passive mode are typically based on PZT, BaTiO_3_, or polyvinylidene fluoride (PVDF) piezoelectric materials [7,27,28,48,66,67,68,69,70]. According to the choice of piezoelectric material, the sensor design or selection is completed by the definition of the fabrication technology and the dimension/shape that must accomplish several system level target parameters, such as:bandwidth;sensitivity/Gain/signal to noise ratio (SNR);input Impedance;input signal dynamic;temperature range;mechanical features: Stress/Strain/Brittleness/Flexible/Stretchable;bonding/Embedding;electrical connection/wiring; and,cost.

Typically, single element sensors have planar dimensions in the order of several millimeters and they operate in non-resonant mode; these conditions lead to an almost isotropic (omnidirectional) sensitivity to UGW and broadband frequency response (e.g., 20 kHz–1 MHz), so they are versatile sensors for many applications, as they cover a large range of the *fxd* product of the phase velocity diagram (see Section 2.1). On the contrary, these broadband sensors are not UGW mode selective and, as pointed out in Section 2, the overlapping of different modes requires clever signal processing to extract information on impact position.

For example, a comparison of different type of sensors can be made by observing three different and common sensors technology for UGW detection (see Figure 4). By comparing the electrical, mechanical, and piezoelectric characteristics of these three types of materials, it is quite straightforward to select the most appropriate sensor technology for the targeted application.

Table 1 reports the characteristics of these three sensors shown in Figure 4.

By the analysis of Table 1, for three sensors having comparable area, the difference in capacitance can be pointed out, which is a relevant parameter for the electronic design (see Section 4). In addition, the different acoustic properties imply different performance for the acoustic matching with different materials, like metal or CFRP, with consequences on the sensor sensitivity: piezoceramic materials are well matched with metals, and piezocomposite and piezopolymers are better suited for plastic composites.

Several types of commercial and customized sensors can be compared according to the list of nine points that are reported above. For example, Wu et al. [68] compared the commercial Accellent Smart Layer^®^ sensors arranged in a SMART Layer (SL) with PZT flexible ultrasonic transducers (FUT) that were fabricated with sol-gel process in order to achieve a large bandwidth for inspection of materials with large thickness surface waves (3–6 MHz) or for NDI of small *kxd* products of laminate materials with UGW (300–600 kHz). An interesting publication regarding the state of art for in service application of commercial transducers for SHM in aerostructures is available [71].

Qi et al. [63] compared PVDF film and PZT patch sensors for the impact monitoring of low velocity impacts in smart aggregates, and concluded that there are relative merits for both materials. Jia [64] analyzed the dynamic response of embedded PVDF sensors at different impact velocities (see Section 2.2). Qi et al. [72] compared PVDF film and PZT patch sensors for impact monitoring of low velocity impacts in smart aggregates and the conclusive remarks is that there are relative merits for both materials. Jia [73] analyzed the dynamic response of embedded PVDF sensors at different impact velocity (see Section 2.2).

Recently, the research has been moving toward new sensors, and there are important novelties in the research of functional materials with enhanced piezoelectric properties: an example published recently by Han et al. [74] is the development of highly sensitive impact sensor based on a PVDF-TrFE/Nano-ZnO composite thin film. The percentage of doping of PVDF TrFe Copolymer with ZnO increases the sensor sensitivity and the dielectric constant. That paper also reports preliminary results on signal acquisition for different impacts. Another approach was proposed by Capsal et al. [75] by the technology development of a flexible, light weight, and low-cost electroactive coating obtained by the dispersion of BaTiO_3_ submicron particles on a in a polyurethane matrix; the experimental set up was demonstrated to detect impacts on an aircraft structure in real time. Finally, Kwon et al. published a recent study on piezoresistive properties of silicon carbide (SiC) [76]: an SiC fiber sensor network has been embedded in a composite structure for low-velocity impact localization on a composite structure. The SiC fibers have the potential to reduce the mechanical discontinuities that were introduced by the sensing elements that is a critical point for the embedment of many types of piezoelectric elements. Another innovative approach that was introduced in [77] is the adoption of nanotechnologies for embedding carbon nanotubes (CNT) into composite materials and the analysis of electrical resistance variation for high and low energy impacts is shown. The introduction of new materials for sensing impacts and damage monitoring is a new fertile field for the research, and the advantages and disadvantages with respect to common piezoelectric sensors will be clear when such devices will be more mature by moving from laboratory to real-field tests.

### 3.2. Multifunctional Sensors Based on Piezopolymer Film Material

The possibility of using the same device operating in passive mode for impact monitoring and for damage detection and localization in active mode is an important advantage of simplifying the SHM system complexity. Moreover, adding sensing capabilities to the same device as temperature or strain measurements lead to a new type of devices that are called “multifunctional” sensors. For example, the data that are obtained from these devices are usefully processed by clever algorithms to compensate for variation of the UGW propagation and physical sensor properties due to thermal drift (see Section 2.3). In addition, the UGW mode selection for the damage evaluation is another useful requirement to have in a transducer. In this section, we will explore the concept of a multifunctional sensor that is based on interdigital transducers (IDTs). IDTs for guided Lamb wave offer the advantage over single element transducers (see Figure 4) in the selection of Lamb wave mode for a given material by the definition of the kxd product (see Figure 2); in this regard, they can be considered as narrow band devices. IDTs for guided Lamb wave applications are created by a sheet (or thin plate) of piezoelectric material equipped with electrodes on the opposite surfaces: at least one side must host two sets of interleaved comb electrodes with separate connections, while the other may present a ground plane, another pattern of electrodes. Figure 5 shows a common exploded view of an IDT, where the geometrical parameters are also defined. The transducer has one side coupled to the ultrasonic wave guiding medium (a plate-like structure). The two sets of comb electrodes are generally assumed to operate with 180°-out-of-phase signals (both in transmission and reception), such that the transducer provides geometrical wavelength selectivity when attached to the surface of a plate-like waveguide. The IDTs that are made by piezopolymer film, like PVDF, have a unique advantage with respect to ceramic of flexibility and conformability to non-planar surfaces, but, according to Table 1, their limits on the temperature range as well as different sensitivity must be well understood and the investigation results are reported in the following sections.

The IDTs developed by our group present a difference with those that were published by other research teams, in that they are manufactured via laser etching, starting from metal-coated—usually with Pt-Au, or Cr-Au alloys—poled PVDF sheets. Because PVDF is mostly transparent to the laser beam, it does not heat up considerably during the etching process, and the laser also passes through the polymer etching the back-side metallization. Therefore, the process results in having an identical electrode pattern on both sides of the PVDF. The possibility of reproducing, with a quick process (tenth of seconds), an arbitrary pattern on the metal coating of the piezo-polymer film by laser ablation represents an enabling technology for including different sensing elements on the same film and reduces the production costs of multifunctional sensors.

For this purpose, two additional sensory patterns have been etched alongside the IDT electrodes on the same piezo-polymer film device: a 1/4” circular element (impact passive sensor) and a resistive temperature device (RTD). The picture that is reported in Figure 6 illustrates the three patterns for the multifunctional sensor alongside the dimensional drawing.

### 3.3. Comparison of Piezoelectric PVDF and PZT Sensors Sensitivity for Impact Detection

In the previous section, we reported the design and fabrication of a circular sensor that was integrated in the same IDT device with the aim to capture impact generated Lamb wave signals propagating from any direction with respect to the sensor center. Some companies have specialized in providing patch piezoelectric sensors with characteristics that are suitable for acoustic source localization, and off the shelf devices are available from Acellent and Physik Instrumente. Specifically, in our design, the circular PVDF sensor has a diameter of 6.5 mm, similar to Acellent’s SML-SP-1/4-PZT sensor (1/4”, or 6.35 mm) (see Figure 4).

The sensitivity of the circular piezoelectric element as a receiver was assessed by comparing it to a PZT device of a similar active area (see Figure 4), the Physik Instrumente P-876.SP1. These two sensors were taped side-by-side to an aluminum plate 1.2 mm thick, with a third transducer used as transmitter and placed at distance of 200 mm from both. A Morlet wavelet centered at 250 kHz was transmitted and received using the same pre-amplifier for both sensors: an instrumentation amplifier (INA) providing a voltage gain of 78 dB at 250 kHz. The excitation wavelet and the acquired traces are plotted in Figure 7a,b, respectively.

The plot shows that, as expected from the piezoelectric properties of the materials, the circular element sensitivity is lower than the PZT device. However, such a wide difference may not be a problem in impact detection applications, where the signals tend to be rather large, as reported in [73], for different impact velocities. In some cases, the large input voltage at the preamplifier input exceeds the rail-to-rail input and saturates the output with a consequent loss of information of the impact event. In conclusion, the different sensitivity of the two piezoelectric materials is not a limiting factor for the choice between the two. There are other differences between that must be considered for the choice of the sensor technology as temperature. In the following section, we analyze the operating temperature range of PVDF piezo films, being limited with respect piezoceramic and piezocomposites (see Table 1).

### 3.4. Operating Temperature Range Estimation of Piezopolymer Sensors

In this section, we report the assessment of the temperature operational limits of the PVDF material used in harsh environments (e.g., aerospace). The authors carried out some measurements at cryogenic temperatures (up to −80 °C) and at high temperatures (up to +50 °C) while using a piezopolymer sensor pair in pitch-catch mode, realized with P (VDF-TrFE) copolymer film.

A series of cryogenic treatment tests of the P(VDF-TrFE) film sensors were conducted at the following temperatures: −20 °C, −40 °C, −60 °C, and −80 °C.

The conditioning procedure consisted of the following steps:Inserting the sample into the steel tube housing (see Figure 8 left).Immersion of the sample in the cryogenic chamber remaining above the liquid nitrogen level.Time to reach the desired temperature (from 20 to 40 min).Test duration time 20 min.Sample recovery time up to room temperature 15–30 min.Test the sample on reference aluminum laminate supplied by TAS-I (see Figure 8, right), using sample IDT #1 as transmitter and IDT #2 as receiver.

First, we attached the sensor pair (IDT #1 and IDT #2) to an aluminum laminate with a bi-adhesive tape at a certain distance in a pitch-catch configuration, and we recorded the ultrasonic signal collected to the receiver transducer before the treatment. Subsequently, we removed the receiver and treated it at the cryogenic temperatures. After the treatment, we repositioned the receiver on the plate and again recorded the signal received. When comparing the collected signal to the receiver after the temperature treatments, we pointed out that no variation in terms of signal amplitude has been recorded for all cryogenic testing temperatures.

Another test has been carried out at temperatures up to +50 °C by heating a pair of sensors that were attached with a bi-adhesive tape (furnished by Eurocel—SICAD group) to an aluminum plate in a climatic chamber for about one hour. The detailed description of the testing procedure is reported in the following:Setting of the desired temperature by remote programming of the air conditioning system with Peltier cell.Wait for the time to reach the desired temperature equal to 15 min.Test duration time 20 min.The acquisition of the signal on the IDT # 1 sensor, using the IDT # 2 as transmitter.

Subsequently, we recorded the ultrasonic signal that was collected to the receiver transducer before the treatment, and we recorded the same signal after reaching the temperature of +50 °C. Again, when comparing the collected signal to the receiver before and after the temperatures’ treatment, we pointed out that no variation in terms of signal amplitude has been recorded. After these tests, we concluded that this type of material could be used certainly down to −80 °C and up to +50 °C without degradation in its piezoelectric properties. The thermal properties are also relevant for the permanent bonding of piezopolymer sensors on the structure by epoxies that often require curing temperature up to +60 °C.

### 3.5. Advanced Technologies for Piezoelectric Sensors in SHM Systems

The main piezoelectric materials analyzed in Section 3.1 have been used to design different types of sensors and transducers in the last two decades with the scope to be integrated with the target structure. In this section, we will review the developments of more advanced sensors and transducers designed for achieving different characteristics:embedded sensors with the structure,Lamb wave mode selection, andarray configuration.

Sensors and transducers are often combined for passive and active mode operation. Lehman et al. [53] reported the advantages of a piezocomposite transducer made by PZT fibers demonstrating the possibility to integrate such a transducer in an aircraft wing. This early paper introduced the concept of sensor node with electronic integration and connection to a base station; Figure 9 shows a graphical description of this system configuration. The same paper also addressed the advantages and disadvantages of the removable sensors with adhesive tape bonding with respect to permanently bonded sensors in composite structures; this problem is often found when a prototype system is to be tested in a laboratory before final testing on the final structure. It is worth noticing that this type of sensor was also tested for impact detection based on the observation of a dispersive A_0_ mode that was generated in a CFRP plate.

#### 3.5.1. Sensors Embedding

Another issue for sensors is the embedding in the structure to ensure durability for service in harsh environmental conditions. Bellan et al. carried out an example of the embedding PVDF IDTs for composite CRFP materials [78], but no easy solution for connections and wiring of the piezoelectric film was provided. Following these early works, an innovative approach that was based on bioinspired sensors was proposed by Ghoshal et al. [79] with a ribbon of PZT element array. Recently, the concept of “smart-skin” (SS) of bioinspired embedded sensors was developed with several advantages in the installation and the simplified task for signal acquisition and processing [80]. Another interesting approach for “aircraft smart composite skin” (ASCS) was proposed in [81], with the investigation of efficient ways to connect in series and/or parallel a large number of PZT sensors with front end electronics; a signal processing strategy for converting analog information to digital sequences was also a main result towards simplifying the embedded signal processing.

Another innovation on sensor technology was stimulated by the installation of stretchable sensor networks on structures that were subjected to large mechanical deformation/strain under mechanical loading (e.g., Composite Overwrapped Pressure Vessel (COPV)) [82]. The concept of flexible sensors has been further investigated in [83], with “bioinspired stretchable sensors” (BSS) presenting multifunctional capabilities; a screen-printed PZT technology on a substrate flexible electronics is envisaged as an enabling technology for integration of SHM system with the monitored mechanical component. Another interesting review of novel EMI methods for integrating piezoelectric sensors in a concrete structure or in a transportation vehicle is reported in [84]; these two different target installations both imply operation in a harsh environment; therefore, the sensor protection by adding an additional layer or by embedding is a key point to ensure the durability of the sensors and the system functionality. Hu et al. proposed another recent work regarding the application of stretchable sensors for AE location [85], where an array of 10 × 10 PZT elements encapsulated in silicon elastomer layers have been developed and preliminary tests on non-planar 3D surfaces are reported.

#### 3.5.2. Lamb Wave Mode Selection

For the Lamb wave mode selection, a suitable transducer structure is the IDT, as reported in Section 3.2. PVDF IDT type of transducers were the first proposed by Monkhouse et al. [66] to generate Lamb waves in structure and he following works by Capineri et al. [71] and Mamishev et al. [72] have developed the fabrication technology, while the analysis of electrodes shape for tunable transducers is reported by Lissenden [57]. The latter characteristic is fundamental for the mode selection that, in many cases, simplifies the interpretation of the signal information. An extensive review of the IDT technology is provided by Mańka et al. [86] and Stepinksy et al. [70] for tunable IDT realized with piezoelectric micro-fiber-composites (MFCs). Arrays of IDT that were employed in passive mode for impact detection have been experimented in an integrated SHM monitoring system for pressurized tanks by Bulletti et al. [69], but the location accuracy needed was limited by the anisotropic sensitivity response of the IDT, as demonstrated by Lugostova et al. [87]. Moreover, the evolution of IDT used in both passive and active mode is the array configuration where each pair of finger electrodes can be connected independently to a channel of the AFE, which allows for driving or receiving signals with different time delay and gain to improve the Lamb mode selection and apply signal apodization, as shown by Bulletti et al. [88].

Because the anisotropic response of the IDTs is a limiting factor when used as impact sensors, several works have investigated this design issue from the theoretical point of view, as Thompson et al. [89] and Wang et al. [90] and by the experimental works of Mańka et al. [91] and Lugostova et al. [87,92]. The multifunctional sensor solution with a circular piezoelectric element included in the same device with an IDT and a RTD sensor can overcome the problem of isotropic and broadband impact sensing without adding complexity and cost to the system, as shown in Section 3.4 (see Giannelli et al. [93]). In this regard, a complete review of the SHM sensors technologies and systems was recently published by Qing et al. [48], where a network of multifunctional sensors for environmental adaptivity is proposed: EMI, UGW, RTD, and strain data can be used and correlated to minimize the influence of variable operating conditions.

The concept of Ultrasonic Guided Mode (UGM) selection by an IDT tunable transducer have been expanded by studying different electrodes geometries, like the spiral transducer developed by De Marchi et al. [94]: in that paper, the synthesis of directivity is presented and it can be usefully adopted for the definition of the sensors layout and number of sensors/transducers to be installed on a defined structure; moreover, that paper also indicates a suitable signal processing strategy based on DTOA information for considering the spiral based patterned geometry. Another type of electrode patterning has been studied as the annular shaped IDT designed for SHM application that was published by Koduru et al. [95] and Gao et al. [96]. This solution has been recently implemented with screen printed technology by Salowitz et al. [97].

#### 3.5.3. Array Configuration

In general, sensor networks are installed on the structure to have an optimal area coverage. An alternative solution is the installation of an array of transducers for implementing the scanning of the area by electronic beam steering in transmission and receiving mode. The latter is also of interest for the implementation of algorithms for the estimation of the direction of arrival of a Lamb wave that was generated by an acoustic source. The programmable beam direction of a transducer emission and reception can be obtained by the well-known phased array solution common in the NDT and medical ultrasound echographic instruments, equipped with integrated analog-digital electronics to achieve real-time beam steering. Generally, high spatial resolution imaging is obtained for the ROI selected on a portion of the plate-like structures, which must be reachable by a line of sight from the phased array without obstacles (inserts, stiffeners, and bolts) in between. The SHM that is based on phased array implies a higher cost, higher power consumption, and it is not scalable with the dimensions and shape of the structure. Giurgiutiu et al. recently published important developments [98] with arrays based on piezoelectric wafer active sensors (PWAS) and, more recently, by Ren et al. [57,99], with 16 elements PVDF arrays operating in a broad bandwidth (0.2–3 MHz). An example of an embedded instrument that is capable is programming a phased array by remote connection is the Pamela project developed by Aranguren et al. [100]: an embedded electronic instrument with Field Programmable Gate Array (FPGA) can be programmed for specific signal processing of data that were acquired by a 16 piezoelectric element phased array.

## 4. Influence of Front-End Electronics on Impact Detection and Localization

In the previous sections, we have described the importance of the choice of the sensor technology and configuration for passive impact sensing, while, in this section, we will address and explain other important issues for the analog front-end design: impedance matching, input signal dynamic, bandwidth, distortion, and power supply. The role of electrical impedance matching is crucial in SHM integrated system design, as the operating bandwidth is continuously increasing and different types of AE sensors are in use; with this aim, Rathod et al. published a recent paper [101]. Poor electronic design can lead to a loss of information on the impact event, as reported by Qing [48], where several approaches are presented to process the signals generated by a set of sensors; other relevant works for the electronic design developments of sensors network are the Match-x project [102] and the work of Ferin [103]. A useful reference paper for AFE designers was published by Beatie [104], where an analysis of important electronics characteristics of the AFE and their influence on the overall impact detection performance is reported. Today, Analog to Digital Converters (ADCs) can acquire at a sampling frequency (F_sampl_) of 50 MHz with a 16-bit resolution and at low power with 3.3 V voltage power supply. Such a resolution implies a 90 dB dynamic at the ADC. This large input signal dynamic range is useful to preserve signal integrity when both low and high velocity impacts must be monitored. The choice of the sampling frequency is important to avoid oversampling nuisance in automatic signal processing schemes and high data rate transmission from sensor node, as noticed in the work of Ebrahimkhanlou et al. [105]. Typically, for a broadband SHM system, a maximum analog bandwidth of 1 MHz is required, which leads to a minimum frequency sampling of 5 MHz when considering a five-fold factor; at this sampling rate, the new ADC technologies have a low power consumption.

### 4.1. Programmable Single Channel Front-End Electronics for Signal Conditioning

In this section, we will explain the advantages of designing or using a programmable electronic for interfacing piezoelectric sensors with different impedance and sensitivity, and we will review the main design concepts. Figure 9 shows the main electronic components of a programmable single channel AFE, and we include a numerical example for the evaluation of performance; the list of the main components is reported, as follows:(1)A low noise amplifier (LNA) with a fixed open loop voltage gain (typically 10 dB) and programmable feed-back impedance to match the sensor impedance bandwidth equal or larger than the sensor (e.g., 50 kHz–1 MHz). For example, we can assume a Noise Figure (NF) better than 5 dB and input equivalent noise density 0.6 nV/√Hz.(2)A programmable Variable Gain Amplifier (VGA) for adjusting the signal amplitude to the input voltage rail of the ADC (e.g., selectable gain −10 dB, +30 dB).(3)A passive anti-aliasing filter (AAF) with an attenuation rate depending on the filter order (typically 6 dB) and a cut-off frequency f_cut-off_ equal to the higher spectral component of the input signal.(4)An ADC with sampling frequency F_s_ selected according to Nyquist criterion and more than 5–20 times the f_cut-off_. The ADC should be selected with a low equivalent noise floor.

With a numerical example, we will illustrate, in this paragraph, the quantification of intrinsic noise for a chain that allows a programmable gain through a VGA. The total voltage gain can be calculated with the reference component values and the max VGA gain of 30 dB:(2)AvTOT(dB)=Av (LNA)+Av(VGA)−Av(AAF)=10+30−6=34 dB or about 50 V/V

For this value, when considering an input dynamic of 3 V that is dictated by the rail of power supply voltage of the ADC, we can manage a signal input that is generated by the sensor with voltage Vs:(3)Vs =3V/50=60 mV

Assuming an equivalent noise density for a 16 bit ADC of Vn (ADC) = 30 nV/√Hz, we can calculate that the equivalent input noise for the maximum AvTOT(dB) that is:(4)Vn_in (ADC)=Vn(ADC)/AvTOT(dB)=30 nV/√Hz/50=0.59 nV/√Hz

This equivalent input noise should be equal or smaller than the intrinsic input noise of the LNA and, in this case, the criterion is satisfied, being 0.6 nV/√Hz.

The setting of the max VGA gain can be changed to adapt the amplification of signals that are generated by a higher energy impact to avoid saturation, for example, a vs. = 200 mV. AvTOT(dB) can now be recalculated by (2) for this case:(5)AvTOT(dB)=3V/0.2V =15 V/V

According to (3), the Vn_in (ADC) increases to the new value:(6)Vn_in (ADC)= Vn(ADC)/AvTOT(dB)=30 nV/√Hz/15=2 nV/√Hz

The new operating condition shows a decreased SNR performance, with the ADC input noise exceeding the LNA noise. Assuming the worst case of the latter example for a bandwidth of B = 1 MHz, the equivalent input noise voltage is:(7)Vn_in_equivalent (B =1 MHz)= Vn_in (ADC) × B =2 nV/√Hz × √1MHz =2 mV

This value needs to be compared with the lower amplitude of the Lamb wave mode signal that can be received for a given sensor sensitivity, especially if a signal processing scheme is based on a threshold method. Low impact velocity impacts often generate fast S_0_ mode signals in the order of tens of microvolts and, in that case, the AAF must be designed to the minimum bandwidth requirements and the voltage gain set to the maximum available in the chain.

This analysis explained by relationships (1)–(7) is useful to demonstrate one of the trade-offs for the design of the AFE when the input signal has large amplitude variations. A good example of this situation is the signal conditioning of an impact signal that is described in Section 3.2, where the generated S_0_ leads the slower A_0_ and the amplitude ratio between the two signals can be a 10-fold factor.

These problems (SNR, gain setting, dynamic) are partially overcome today by using cthe omponent of the shelf (COTS) integrated circuit for AFE, but their characteristics are often optimized for the NDT and medical ultrasound sensors, while, for the SHM, the input voltage levels and bandwidth differ from those fields. Moreover, the integrated devices that include ADC have steady state power consumption that is compatible with power supply units for electronics in a base station (see Figure 8), but such power consumption is rather demanding when the electronic front end is close to the sensor, as for the solution of a battery operated node for a sensor network. Yun in his master thesis [96,106] proposed an electronic solution for impact detection with nodes implementing EMI method where the impact signal triggers a low power comparator that switches on the power supply of the rest of the electronics for acquiring the signal over a defined amount of time. This type of solution alleviates the problem of power supply for continuous monitoring. Thomas et al. [107] demonstrated that a coverage with AE sensors deployed along annular rings installed on a composite tube can produce high quality images of damage by an EMI tomographic method.

The pick-up of environmental noise when broadband sensors are adopted is another electronic design issue. The extrinsic electromagnetic noise that was picked up by the wiring of the sensor to the AFE is an additional source of SNR deterioration unless bulk coaxial cables are used. The differential connection of the sensors is a quite robust solution that mitigates the common mode noise, but this implies the design of special differential amplifiers with high common mode rejection ratio (CMRR) at the operating frequency, as reported by Boukabache et al. [108] and Capineri et al. [109].

### 4.2. Real Time Electronics for Impact Monitoring

In this section, we review the developments on real time electronics for monitoring multiple impacts with multichannel inputs capability, which is a mandatory feature for implementing large sensor network experiments and installation.

From the research point of view is also very important to test the whole SHM with multiple impacts to gather many signals in real time, as shown by Ren et al. [110]. This approach allows, with laboratory experiments, to simulate repetitive impacts at different energy levels and periods to test and optimize the sensor layout and electronic signal conditioning parameters. The multiple impact experiments can be done in the laboratory with programmable mechanical impactors, as reported in [27,50]. This solution is very useful for avoiding time consuming experiments that are based on the pencil-lead break (PLB) tool for the collection of large signal data bases to test advanced algorithms (see Ebrahimkhanlou et al. [105]). Impact detection and positioning is obtained with several sensors (at least three) that were deployed on the structure with a strategy for uniform area coverage and detection sensitivity.

For these reasons, several recent works have proposed real-time electronic platforms with multichannel capabilities to overcome the main limitation that is posed by the common solution of using a general-purpose digital oscilloscope. A real-time electronic platform design for passive and active mode functionalities was published by Capineri et al. [111], while Yuan et al. [112] designed a low-cost signal acquisition system based on sensors tags with local preprocessing capability.

In early works that were published by Ziola [26], the evolution from narrow to broad bandwidth sensors and analog front-end systems was proposed to locate the acoustic source more accurately, as the spatial resolution is improved by using higher frequency UGM. Impact velocity and energy variability generates different modes and, for the calibration tests, are often recommended low energy impacts that were carried out with the PLB as acoustic source, as reported by Wilcox et al. [63]. Gao et al. also discussed the advantages of retrieving information from broadband signals [113].

### 4.3. MEMS Sensors, CMUT, PMUT, and Integration with Electronics

Advances in micromachined electrical mechanical systems (MEMS) over the past two decades have opened up the search for a new class of sensors for AE and SHM. MEMS technology has been very successful in integrating sensors with electronics, especially in achieving low-cost mass production thanks to integrated circuit technologies. Tri-axial capacitive MEMS accelerometers are probably the first example of such an integration process that started in the 80’s and has now achieved important results in multisensory nodes (MOTES), as reported by Glaser et al. [114]. In this section, the focus is on deterministic sensors for SHM and AE based on UGW for both passive and active mode, as introduced in Section 2. The interest in deterministic sensors that are capable of directly producing data useful for the detection and growth of defects has attracted interest in finding alternatives to PZT, Aluminum Nitride (AlN), Zinc Oxide (ZnO), and piezoelectric/piezoresistive UGW devices. Actuation and sensing UGWs by capacitive MEMS is derived by the first study of Haller et al. [115] at Ginzton Laboratory, Stanford University, based on the electrostatic actuation of a thin silicon membrane. At first capacitive MEMS technology was meant for improving airborne ultrasonic transducers, but it immediately revealed the potential application for generating Lamb waves in solid materials (see Yaralioglu et al. [116]); after two decades the recent advancement of capacitive MEMS sensors in the design, fabrication, and integration with electronics can be found in the review paper of B.T. Khury Yakub [117]. Since then, the effort in designing small scale factor Capacitive Micromachined Ultrasonic Transducers (CMUTs) for SHM and AE has been great, and different design and fabrication methods have been proposed in the PhD dissertation of Bradley [118] and, recently, by Butaud et al. [119]. CMUTs are generally designed as resonant devices and the resonant frequency depends on the bias voltage. The front-end electronics for CMUT are generally different from that one that is required for low impedance piezoelectric devices; the essentially capacitive behavior of the sensor impedance requires a custom design of the LNA (see the signal chain in Figure 8). In this regard for testing commercially available CMUTs in laboratory setups, charge amplifiers, suchas CA7/C by Cooknell Electronics Ltd., have been used by Bradley [118] and Butaud et al. [119], while the opportunity to use on-chip integrated multichannel Analog Front-End (AFE) for CMUTs was reported by Savoia et al. [120]; more recently, B.T. Khury-Yakub presented the approach of monolithic integration of a CMUT array with Application Specific Integrated Circuit (ASIC) based on flip-chip bonding [117]. For the detection of Lamb waves, CMUTs still need to be improved in terms of sensitivity and signal to noise ratio with respect to conventional piezoelectric sensors, as reported by Boubenia et al. [121]. MEMS technologies were also applied for designing and fabricating piezoelectric devices. Generally speaking, a piezoelectric MEMS sensor for SHM is based on a resonating silicon microstructure and a thin piezoelectric material layer and it is assembled in a ceramic package. The main advantage is to retain the high electromechanical coupling coefficient of piezoelectric materials with the advantage of a significant reduction in size and weight. The latter are promising features to ease the installation and embedment into the structures. An alternative technology for sensor systems size reduction are Piezo-MEMS. There are two recent works that were published by Ozevin et al. [122] reviewing the advancements of piezo-MEMS operating in the 40–200 kHz frequency range. The reference [123] reported MEMS based on both piezoresistive materials that need to be supplied by constant current sources with temperature compensation; the same review work also describes another type of capacitive sensors for AE that differs from CMUT, as it is based on the change of capacitance in response to a dynamic stimulus that varies the distance of the electrode plates. This principle is well known in capacitive MEMS accelerometers, and it is demonstrated for inplane wave sensing through a differential capacitance sensor for AE applications [124]. That review paper also addresses the main difference between broadband and narrowband devices: while the latter have high sensitivity at the designed resonant frequency with high Q factor, the broadband are more versatile devices, but the sensitivity is not yet comparable with analog bulk piezoelectric sensors. The increase of active area of the piezo MEMS increases the sensitivity, but their footprint gets closer to those of conventional piezoelectric sensors. However, for some applications where high energy impacts generate large amplitude stress waves in the structure, the lower sensitivity of MEMS sensors can be acceptable. Despite these advantages of miniaturization and integration with AFE circuits, these devices lack experimentation in harsh environments or at least in simulated operative conditions for aerospace, automotive, and civil engineering applications. Guemes et al. recently published a review paper that also discusses the additional problems when the sensors are permanently attached to a structure [125]: the reliability of the entire SHM system needs to be studied with more focus in order to demonstrate the sensor and electronics technology for real life applications. Finally, another MEMS technology that is investigated for AE sensor is the piezoelectric micromachined ultrasonic transducer (PMUT), first introduced in the 90′s for ultrasonic applications in the 100 kHz–15 MHz range by Percin et al. [126], Muralt et al. [127], and Bernstein et al. [128]. The main concept for the P-MUT device was the design of a sensor based on laminated structures vibrating in the bending mode by combining the rigidity and strain of beam and plate microstructures. This technology has been also been recently applied for AE sensor and Feng et al. [129] developed a PZT micromachined cantilever-based sensor. The comparison of the new PMUT device with a commercial sensor seems to be promising besides the characteristics (gain, bandwidth, and filtering) of two adopted AFEs should be compared.

## 5. Hardware Developments of Wired and Wireless Sensor Networks (WSNs) for SHM and Validation Tests

From the previous sections, it turns out that, in recent years, the combination of several progresses in sensors and mixed signals low power electronics have introduced a new paradigm for the SHM systems that is the network of sensors nodes, as reported by Farrar et al. [130]. Figure 10 shows a conceptual description of the migration from single distributed sensors on a structure to the sensor network, where, for example, the authors represented a sensor network for monitoring a COPV system. In the same picture are shown the main electronic blocks that are needed to realize a sensor node with active and passive mode operation. The transducer driver (for broadband or narrow band ultrasonic transducers) and the signal conditioning are both controlled by a mixed signal System on Chip (SoC). The connections between nodes and the central unit (see architecture in Figure 10) can be implemented with wired solutions where the power lines for the nodes can sustain a sufficient data rate by using power line communication (PLC) protocols and related chipset. Simplified connection schemes and a low power digital electronic front end has been recently proposed and validated on an aircraft wing by Qiu et al. [81]. The SoC development of a node with passive and active mode operation poses several design issues that are related to the electronic design. The main issues are the power consumption and design of an efficient ultrasonic pulser to gain transduction efficiency in active mode [131]. Local high voltage power amplifiers or pulsers are needed to excite transducers with 10 V to 100 V amplitude excitation signals; the local availability of high voltages is generally obtained with boost DC/DC converters. This type of converter can be realized with SoC solutions, but the integration of passive components (inductors and capacitors) still needs to find a compromise between the size and switching frequency. The dimensions also become critical for the integration into the structure and protection of electronics is needed to guarantee a life-time same as the monitored structure. The cost of wiring is generally high, and the replacement of defective hardware and sensors should be avoided for a time that is comparable to the service life of the facility.

Schubert et al. published one of the first implementation of this paradigm [102] with the Match-X project of the Fraunhofer Institute. The node design and electronic integration with a stack of miniaturized PCBs with embedded PZT transducers that were mounted on a glass-fiber-reinforced-polymer (GFRP) plate is reported. The paper also addressed the requirement of power supply overvoltage protection and detection of failure events that is one important consideration for self-diagnostic of nodes. Lehmann et al. [53] presented, in the same year, the results of validation of the embedded PZT MFC transducers in an aircraft wing. Local processing of the acoustic signatures was demonstrated by the integration of the AFE in the node architecture: the ADC, algorithms for data reduction, and digital communication by a Digital Signal Processor (DSP). Although the adopted solution for data transfer was based on a two wires industrial Controller Area Network (CAN) bus, the authors introduced an expandable feature to open the wireless connection with a Bluetooth module, a key feature for the evolution to a Wireless Sensor Network (WSN). Figure 11 shows the main electronic blocks of a sensor node for a WSN.

### 5.1. Nodes and Modules with Low Power Electronics Solutions with Energy Harvesting

The main evolution for continuous impacts monitoring is the concept of autonomous nodes. In the case of an SHM system, we can observe that environmental operating conditions, like those described in Figure 1, are represented by different types of energy exchanges with the structure. This interaction from the point of view of the impact event capture is seen as a disturbance or noise, but from the point of view of local energy accumulation, can represent an opportunity.

A preliminary work testifying to this evolution was published by Champaigne et al. [132], describing a wirelessly connected SHM system to interface up to four PZT sensors and an AFE that was capable of matching with the characteristics of different types of sensors.

In that paper, low power electronics that were available at that time were adopted to be compatible with charge capacity of a dual AA-cell battery pack to reach an operational time up to 10 total hours. A consideration must be made about the careful choice done for digital electronics, such as the ADC, FPGA, and digital communication, which are typically power-hungry devices. A recent paper that can solve the power demands for continuous monitoring is proposed by Fu et al. [133], and the solution consists of keeping in a sleep mode a section of the digital electronic processing until a detected event switches on the power supply of the data acquisition and processing blocks; Overly et al. published a similar approach with a compact electronic design for a wireless smart sensor node [134]. The latter work used low power chips and self-diagnostic for the detection of PZT elements debonding from an aircraft wing. Another important design issue that is tackled in the paper is the temporal synchronization of data from an impact event that was detected by the WSN; this topic will be expanded in Section 5.2. The design of a WSN with low power budget obtained by the sleep mode operability is presented by Giannì et al. [135]; in particular, the authors analyze the design issues regarding the AFE + ADC noise characteristics and their influence on the errors achievable for impact positioning with a triangulation method.

Ferin et al. [103] presented a new hardware development of a highly versatile energy autonomous acoustic sensor node that is an element of an intelligent wireless network; this node architecture is capable of executing various ultrasonic inspection algorithms. The energy harvester was the conversion from mechanical vibrations into electrical energy stored in a supercapacitor with a high charge capacity/volume ratio. In this paper, the hardware specifications for an automated and remote aircraft ultrasound inspection were considered to be a start point for a product-oriented research. Taking advantage of low power electronics with energy harvesting solutions, the design of a MEMS piezoelectric power module converter with a power density of 6 mW/cm^3^/g^2^ and an output power around 120 μW was presented. To cover the full power supply demands of a sensor node, multiple MEMS power modules can be connected at the expense of an increased volume occupation. The piezoelectric energy harvester system was capable of charging a thin film battery (EFL700A39 from STM—700 μA/h 3.9 V). The topic of energy harvesting is strictly related to the design of autonomous sensor nodes and several review papers for the interested reader as Mateu et [136], Sodano et al. [137], and Trigona et al. [138], and an example of a small scale factor energy harvester device is reported in Figure 12. The authors presented in [127] a prototype system for delivering energy to SHM sensor nodes by microwave wireless energy transmission in the 10 GHz X-band. The energy harvesting for low power WSN with special emphasis on SMH application has also been reviewed by Park et al. [128]. Finally, the outcomes of a recent project that was dedicated to the energy harvesting methods for SHM systems installed on airplanes have been published by Zelenika et al. [129]. In [138] the authors presented a prototype system for delivering energy to SHM sensor nodes by microwave wireless energy transmission in the 10 GHz X-band. The energy harvesting for low power WSN with special emphasis on SMH application has been reviewed also by Park et al. [139]. Finally, the outcomes of a recent project dedicated on the energy harvesting methods for SHM systems installed on airplanes have been published by Zelenika et al. [140].

It is also worth mentioning industrial projects covering the WSN approach for aircraft SHM as proposed by METIS Design company [142] and the European Project “FLite Instrumentation TEst Wireless Sensor” [143]. Smithard et al. presented another kind of sensor network formed by modules that were connected by fiber optics to obtain large immunity from environmental electromagnetic noise in [144]. The Acousto Ultrasonic Structural health monitoring Array Module (AUSAM) project relies on autonomous electronic modules that are designed with off-the-shelf electronic components that interface up to 62 PWAS. These modules can operate in active and passive mode, and they are also equipped with an EMI module; the latter is usefully adopted for checking the reliability of the PWASs. A futuristic vision of the AUSAM module is the transportation and installation on the structure by a drone, with some advantages for maintenance service performance and costs. A similar idea of using drones for EMI technique has been recently reported by Na et al. [84]. The interest of sensor networks for SHM in transportation and civil engineering infrastructures also requires a different approach for system performance evaluation; Ju et al. [145] proposed a simulation of a sensor network for the continuous monitoring of railroads, where fast transportation systems are in service. Sundaram et al. [146] reviewed the advantages of WSN for SHM of large civil engineering structures and pointed out the problem of connection reliability, obstructions to radio links, and, finally, the energy harvesting.

Ren et al. [147] presented a strategy for radio communication of autonomous nodes for impact monitoring of large structures and a preliminary validation on a laboratory mock-up of an air wing is presented. The original solution is the adoption of a multi-channel radio communication on different frequency channels to improve the data transmission capability and the reliability of the WSN. Embedded computational resources in sensors nodes for vibration monitoring has been designed and tested on a laboratory mock up by Testoni et al. [148]; this work shows a node design with volume/weight constraints and low power consumption for implementing a wired sensor network based on PCL. A dramatic gain in volume factor for integrated sensor node is achievable by integrated electronic design. With this approach, each electronic block (see Figure 13) can be optimized for low power consumption (oscillators, PWM, ADC, and wake/sleep-mode circuit) and more important the wireless transmission and power management. For the latter solutions based on low voltage single cell batteries are available connected with buck-boost DC-DC converters. The efficiency of these converters is high at low switching frequencies, but it requires an inductor-capacitor (L-C) tank with large component values, which implies a larger volume. Moreover, the integration of different types of transducers (optical, acoustic, and radiofrequency) on a small-scale can ensure the required average and peak power consumption. Lee et al. reported an example of recent development of the integrated custom electronic with multi-chip connection [149].

Summarizing the results of the works that are examined in this section, we can say that technologies for embedded signal processing, low power signal transmission, and their integration with energy harvester devices are now available and they have mainly been demonstrated with laboratory experiments, and some real life installations are featured in the literature. In the next section we will make a discussion of the issues for a wide spreading of smart nodes for SHM networks.

### 5.2. Toward SHM Sensor Networks with Smart Nodes

From the previous paragraph there is strong interest in moving the SHM system toward sensor networks and, in the following section, we will draw some general comments and challenges for addressing the next steps for new developments. In this section, we discuss the advancements of smart nodes in the perspective of an impact sensing SHM network.

The evaluation of data transfer requirements for a node is one of the topics that is now under development for the research. The reduction of data rate for a “smart-node” requires that some local processing is needed. The data rate reduction is achievable by compressive sensing techniques, as investigated by Mascarenas in [150]. The recent research on this subject also demonstrated the benefits for the autonomous detection and localization of an AE source, as we will explain later in Section 6.

The presence of smart sensor nodes, and a relatively dense interconnection network, can provide some degree of redundancy to the SHM system, where failing sensor nodes will not compromise the operation of the overall system. Of course, the thickening of the interconnection network goes against the minimum-encumbrance policy, which is one of the original goals of the sensor network architecture, but it is a trade-off that should, nonetheless, be considered. From the point of view of harnessing, PLC represents a way to achieve the minimum amount of cabling required to route the sensor network, albeit at the cost of reduced bandwidth. A problem that is deeply ingrained in sensor networks that need to cooperate in the ways described above is how to achieve and maintain inter-node synchronization. Although the topic has not been addressed so far, the problem of synchronization in measurement and control networks is well known and it will be approached starting from the provisions of the Precise Time Protocol (PTP) IEEE 1588 standard that can reach a synchronization accuracy of 0.1 µs wired network connected on ethernet. Such performance is compatible with SHM sensor network design being the UGW signals with the frequency content below 1 MHz and Time of Flight (TOF) in the order of 10 µs–100 µs. This analysis derives from the main requirement that each sensor node needs to be synchronized up to a fraction of the DToA to produce data that are useful for accurate impact positions. The synchronization problem is even more complex for WSNs and the next section will go in some detail of the proposed solutions.

### 5.3. WSN and IoT for SHM

In the last few years, the concept of WSN for SHM has moved on to the Internet of Things (IoT) for SHM. The main advantage of introducing the communication of a WSN for SHM over the Internet comes from the possibility to uniquely identify the data packet generated sensor node and the large bandwidth for data transmission; time correlation is achieved thanks to the accurate synchronization of nodes. In addition, the large storage capacity of the cloud allows for further implementing data interpretation using AI and deep learning for Big Data (BD); some examples of the latter novel development will be reported in the next section.

Tokognon et al. [151] have reviewed the challenges for the design SHM using IoT technologies well to achieve intelligent and reliable WSN for monitoring structures. The authors identify three main blocks to be integrated for this aim:Sensing and data Acquisition Subsystem.Data Management Subsystem: preprocessing methods used to organize raw data that were acquired from sensors and remove the noise before processing; novelty detection, classification, and regression approaches. Among them, novelty detection based on artificial neural networks.Data Access and Retrieval Subsystem.

The requirement of low power communication technology based on the IPv6 assignment of a node is analyzed for battery operated sensors. The work of the ZigBee Alliance has accelerated the expansion of the sensor network and building automation market. From the PHY and MAC layers that are defined in the IEEE 802.15.4 standards, Zigbee considers the networking and services layer, through the full application layer. ZigBee PRO was specifically developed for device-to-device communication in an IoT context.

Unfortunately, WSN based on IEEE 802.15.4/ZigBee do not currently support IP, mainly due to the small length of packets that are used in IEEE 802.15.4. Therefore, most of the solutions proposed consist of using IP proxy or gateways. A network configuration strategy for WSN configuration with sink nodes at the edge of the network, also called border routers, with IP protocol connection over the Internet is presented in the paper by Tokognon et al. [151]. From the sink nodes, data can be transferred with JavaScript object notation (JSON) to a Web server, where a large storage capacity is commonly available.

Moreover, the Internet Engineering Task Force (IETF) defined the 6LoWPAN standard (RFC 4944) to allow the use of IPv6 packets over IEEE802.15.4 networks. The new compressed IP headers resolve the packets size issues and the fragmentation mechanism to transmit IP packets over IEEE802.15.4 networks. IETF also started a working group to evaluate the appropriate routing protocols for low-power (RPL) and lossy networks.

The node synchronization is another challenge for a distributed IoT, as stated in the previous section. Scuro et al. [152] published a paper that was devoted to this problem, and a solution was proposed with each node equipped with a clock; the nodesexchange synchronization messages to evaluate the frequency and the offset of their clock with respect to the one taken as a reference (master) or with respect to its neighbor sensor node. This solution implies an additional overhead, since extra messages and re-synchronization periods are required.

In the same structure, local area networks with routers that give priority to the transmission of the synchronization messages, or that compensate for the transmission delay, can be deployed. In these cases, a synchronization accuracy in the order of microseconds is still achievable. In fact, for the SHM system, the typical accuracy that is needed between the node is in the range [0.6, 9.0] μs. Muttillo et al. [153] presented a solution for structural monitoring with digital accelerometers ADXL355 with high resolution that was connected to hardware for IoT connection. A high synchronization between the sensors was implemented to preserve such performance.

Finally, Abdelgawad et al. [142] and Mahmud et al. [143] presented examples of prototype architectures for WSN nodes that were connected on ethernet based on Raspberry Pi. Besides the power consumption of these design was a neglected factor, the two systems were successfully demonstrated for SHM in a laboratory. Finally, an example of prototype architectures for WSN nodes connected on ethernet based on Raspberry Pi have been presented by Abdelgawad et al. [154] and Mahmud et al. [155]. Although the power consumption of these design was a neglected factor, the two systems were successfully demonstrated for SHM in laboratory.

## 6. Artificial Intelligence and Machine Learning

The previous sections pointed out how embedded sensors with low power electronics in a sensor node enable SHM monitoring networks on IoT for large and complex structures. This new paradigm also brings large data collection and data interpretation challenges. In this section, we discuss the recent approaches that were based on BD and Artificial Intelligence (AI), and we then complete the review of all SHM system components that are shown in Figure 1.

One of the early papers on this subject was published by Farrar and Worden [5]. The authors pioneered this subject with the introduction of the machine learning/statistical pattern recognition paradigm for SHM. Since then, in the last decades, remarkable developments have been reported.

Worden et al. [156] analyzed the non-stationary properties of the Lamb waves used in SHM and how the machine learning approach can solve the operator-based data interpretation, which is a time consuming task, especially for networks with large number of sensors.

As said above, BD can potentially enable the automatic classification of defects, but the reduction of input data remains a goal in simplifying the design of the processing task, as proposed by Bao et al. [157]. The application of compressive sampling of sensors signals is a useful strategy and, in particular for Lamb waves, it is worth mentioning the work of Bao et al. [157] where a CNN was trained with experimental data.

Yuhan et al. [158] observed that, in many practical situations, the data are limited to a small period of monitoring time and generated by a specific part of a complex structure, and this limits the performance that is achievable with ML. That work analyzed a possible solution based on physics-informed learning, which integrate information derived from physics-based model into the learning process. Examples of the physics-informed Deep Learning (DL) approach applied for low-velocity impact diagnosis are reported. For this aim, a pipeline consisting of a unified CNN-RNN network architecture for spatial-temporal analysis of the impact generated wavefield was developed. The knowledge type of physical principles was based on classic Huygens principle, time-reversal methods Fink et al. [38] and simulated data based on a dispersive propagation model of generated waves from impacts. This knowledge was introduced in the CNN network of the data processing pipeline and it helps to define a vector feature for the learning and classification. Hesser et al. [159] investigated the autonomous detection of defects in plate-like metal panels with an ANN that was trained by signals acquired by four commercial sensors (PIC255 from PI Ceramic) with 1 MHz sampling rate and 16 bit resolution ADC. The experimental data set was generated by a free falling ball impact at low velocity (about 1 m/s) that are converted in large amplitude, low phase velocity A_0_ mode Lamb waves. This approach demonstrates that the achievable spatial accuracy is on the order of the wavelength that corresponds to the main A_0_ received mode with frequency content well below 100 kHz. Another paper following the work of Hesser was published by Mariani et al. [160], where the autonomous defects classification is explored with a CNN approach that overcomes the limitations of extensive baseline data archives.

Sun et al. [161] have reviewed the framework for the development of damage detection in civil engineering infrastructures (bridges), where big data can be acquired in real time and artificial intelligence strategies need to be adopted.

An interesting approach that is based on data driven models is the application of DL with ANN to directly input raw data from a limited number of sensors for impact localization and characterization has been published by Ebrahimkhanlou et al. [105]. In that paper, a deep network is trained on simulated and experimental data sets with signals received by a very small number of sensors (from 1 to 4) covering the area of a test aluminum panel equal to 500 mm × 500 mm. The single sensor solution is certainly attractive from the point of view of cabling and costs, but for the system reliability, a certain degree of redundancy is necessary by increasing the number of sensors, which also improves the accuracy of impact area estimate and impact characterization.

Another example in the literature is from Melville et al. [162]; the authors reported the investigation of Lamb waves that were generated in an aluminum laminate by piezoelectric transducers, although they used a SLDV to acquire images of the full wavefield, and then used a Convolutanional Neural Network (CNN) for the interpretation. Finally, we observe that the DL approach is capable of exploiting information from signals that were acquired over a long-time interval, where multimodal dispersion and reverberations (multipath) effects are present.

## 7. Conclusions

The paper examines recent developments in integrated SHM sensors and systems for impact detection. The design of advanced SHM systems for impact monitoring benefits from recent advances in UGW modeling, sensor materials for MEMS solutions and interface electronics, signal processing algorithms for real-time applications, sensors for WSN and IOT, and data processing with AI and Big Data.

In the first part of the work, the characteristics of the UGW modes that are generated by the impacts are discussed with the differentiation of low and high speed impacts and their attenuation. The main concepts of this physical background are reported, because they indicate the different characteristics (amplitude, spectral content, and modes velocity dispersion) of the signals that must be processed by the front-end electronics. Subsequently, the characteristics of the most common wideband patch type piezoelectric sensors (PWAS) with narrowband IDT used for Lamb wave mode selection are compared. The introduction of new piezoelectric materials (Carbon Nano Tubes, Microfiber Composites) for MEMS sensors for detecting impact signals is more recent, but promising results have been reported; CMUT and PMUT devices also have a good perspective to be used in SHM for their inherent advantage of the electronics integration. The paper also addresses the design issues for front-end electronics that must match sensor characteristics and impact signals with different energy and operating in the bandwidth 50 kHz–1 MHz. Particular attention to the on-site environmental factors (e.g., thermal excursion, deformation, and vibrations) were also discussed, because they influence the choice of the sensor technology; for the compensation of environmental factors, the research trend is the design of multifunctional sensor nodes and ad-hoc algorithms. The document also shows examples of real SHM installations with operability in passive mode and active mode for damage assessment.

A new emerging technology for reducing the complexity of wired sensor networks is the adoption of a “smart skin” with stretchable/flexible piezoelectric sensors. The anlaysis of several papers on this topic indicates that a trade-off can be achieved between the number of sensors installed and the coverage of the entire area under test. The first part of the review concludes with the description of recent developments on integrated or embedded electronic systems with hardware system on chip with small footprint design.

In the second part of the review, advances in the sensor technology with low-power mixed-signal electronics are reported, as they have changed the architectural design of SHM systems by introducing the concept of “autonomous intelligent nodes”. These types of devices have a microprocessor on board, different types of sensors, wireless communication, are locally powered, and are low cost. Autonomous sensor nodes will also use MEMS devices for energy harvesting in the future, with a power conversion capability above 100 µW and high power density. Wireless communication of a node is now more common, as reliable communication over different frequency channels has been demonstrated. The main reasons for the introduction of the WSN for SHM on the Internet derive from the ability to identify the data packets transmitted by a node and the exploitation of synchronization techniques with latency better than 1 µs to correlate the sampled signals that are generated by an event of impact. The latency in communication between WSN nodes affects the differential time of arrival error, which is the basic data used for solving the impact positioning problem. We can summarize that, in the future, it will be increasingly common to monitor large facilities with sensor networks on the IoT due to the integration of sensors, low-power analog and digital electronics, and efficient wireless communication.

For the off-site components of the SHM system, the document introduces, in the last section, the new challenge of interpreting the impact event on complex structures by collecting large data. Because the complexity of the problem is high, several promising works that are based on Big Data (BD) and Artificial Intelligence (AI) show that the localization of an impact is obtainable with errors being comparable to deterministic algorithms only applicable for simple structures.

An outlook of the future for integrated Structural Health Monitoring systems is reported according to the market growth of the main system components that were reviewed in this paper.

Piezoelectric sensors and transducers: the piezo-MEMS are gaining a share of the market with respect to bulk devices that are holding the big share. The compound annual growth rate (CAGR) for the period 2018–2024 of piezo-MEMS is 15.3%, while, for bulk MEMS, is 12.3%. From this outlook, the devices will be smaller and cheaper, and with lower power consumptionSystem on Chip: the integration of the mixed signal electronics in a single package will benefit of the technological developments of SoCs for automotive and consumer markets with CAGR of 8.4% in the period 2017–2025. The electronics that are realized in a compact scale with single package will be a crucial advantage for the connection in proximity of the sensor/s with multifunctional capabilities for environmental monitoring.Energy Harvesting devices: the global market CAGR for the period 2020–2025 is estimated at 8.4%, and this is a key factor for installing self-powered sensors in installations where the power cable infrastructure is expensive.Wireless Sensor Networks: the CAGR for WSN is industry in the period 2017–2025 is 10.7%. This growth is certainly supported by IOT for Industry 4.0, and the main advantages rely on low-power communications with a data transfer rate compatible with the application to SHM. The programmable configuration of the sensor network is one of the main advantages, especially in applications where a different number of sensors and their position can be optimized during service.Artificial Intelligence Processors: as sensors nodes of SHM plants increase to reach hundreds or thousands of units, the data processing becomes difficult without the support of AI. The electronic market forecast reports a tremendous CAGR for AI application processors of 46% in the period of 2017–2023.

All of these market driven technologies, together with advancement in sensors materials (flexible layers for smart skins), will lead to a rapid evolution of integrated SHM systems.

Overall, the authors of this paper have set themselves the goal of providing a useful reference for readers that are interested in the design, use, and development of on-site and off-site components of advanced ultrasonic wave guided SHM systems.

## Figures and Tables

**Figure 1 sensors-21-02929-f001:**
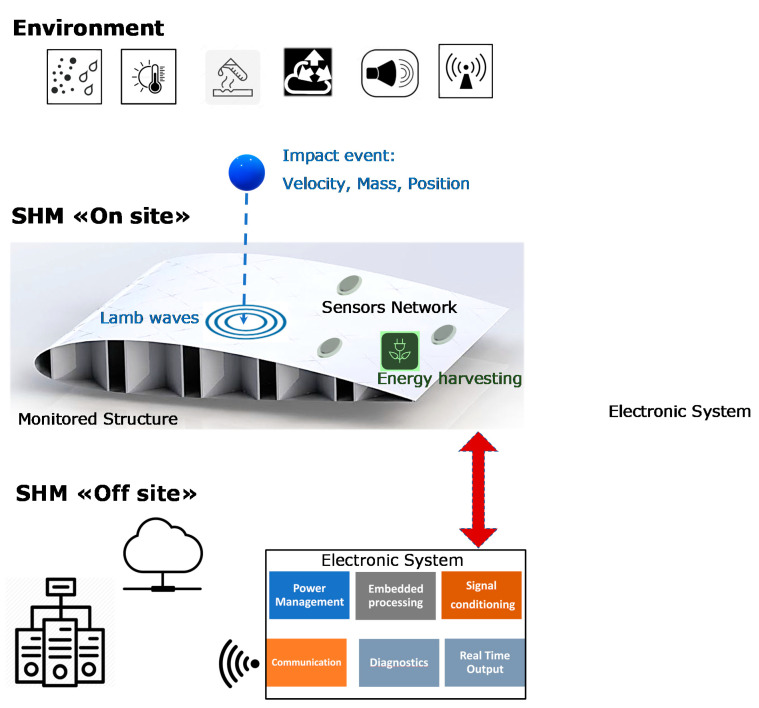
Graphical representation of an advanced structural health monitoring (SHM) system for impact monitoring. (**Top**) Environmental conditions (dust, moisture, temperature, pressure, vibrations, electromagnetic interference) and impact events characterized by the object mass, velocity, shape and dimensions. (Centre) On-site components of the SHM system subjected to environment conditions installed on the monitored structure (e.g., a section of a composite airplane wing). (**Bottom**) Off-site components installed remotely and connected to the sensors network; the Electronic System can operate in a protected environment (e.g., inside airplane fuselage) with real-time processing capability. Off-line signal/data processing based on big data archive with workstations connected to the web for software applications of Artificial Intelligence/Machine Learning (AI/ML) and prognostics.

**Figure 2 sensors-21-02929-f002:**
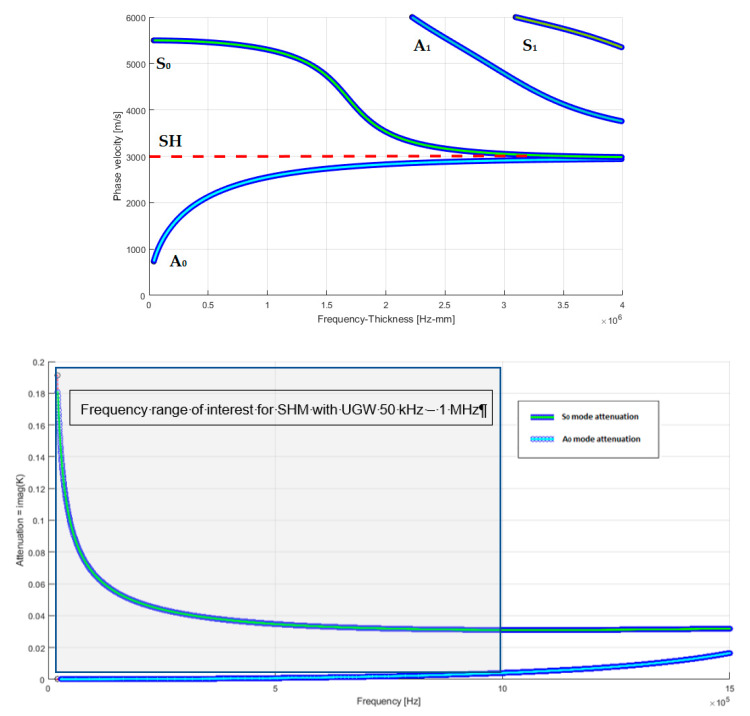
(**Top**) Simulated dispersion curves of phase velocity for low order modes Symmetric (S_0_), Antisymmetric (A_0_) and Shear Horizontal (SH) in an aluminum plate as function of the frequency × thickness product (MHz × mm). The diagram shows that higher order modes (A_1_, S_1_, etc.) are generated well above the value of 1.5 MHz × mm. (**Bottom**) Frequency dependent attenuation of Symmetric (S_0_) and Antisymmetric (A_0_) modes calculated as imaginary part of the complex wavenumber K for an aluminum plate 1.4 mm thick.

**Figure 3 sensors-21-02929-f003:**
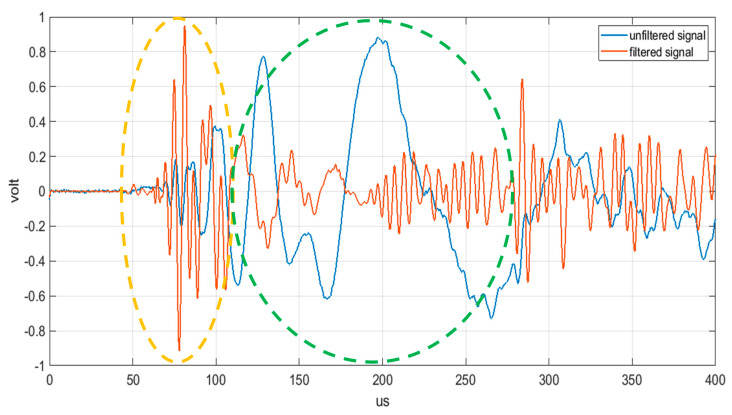
Ultrasonic signals generated by a low-velocity impact (about 3 m/s) in blue color, and the same signal filtered by an analogic low-pass filter with a cut-off frequency of 400 kHz in red color. The dotted green circle represents the portion of the signal relative to the A_0_ mode; the dotted yellow circle represents the portion of the signal relative to the S_0_ mode.

**Figure 4 sensors-21-02929-f004:**
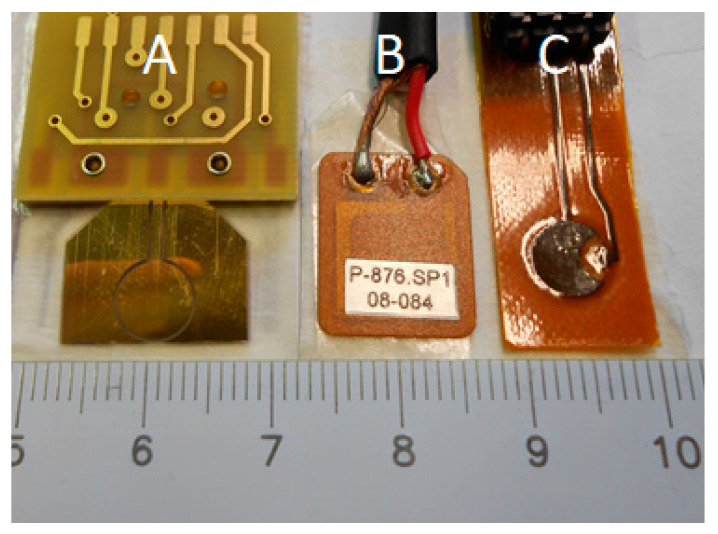
Example of three different type of piezoelectric sensor for SHM: (**A**) circular polyvinylidene fluoride (PVDF) sensor made with bioriented PVDF film furnished by Precision Acoustics, (**B**) BaTiO_3_ piezocomposite, model DuraAct produced by Physik Instrumente, and (**C**) piezoelectric wafer active sensors (PWAS), model SML-SP produced by Acellent.

**Figure 5 sensors-21-02929-f005:**
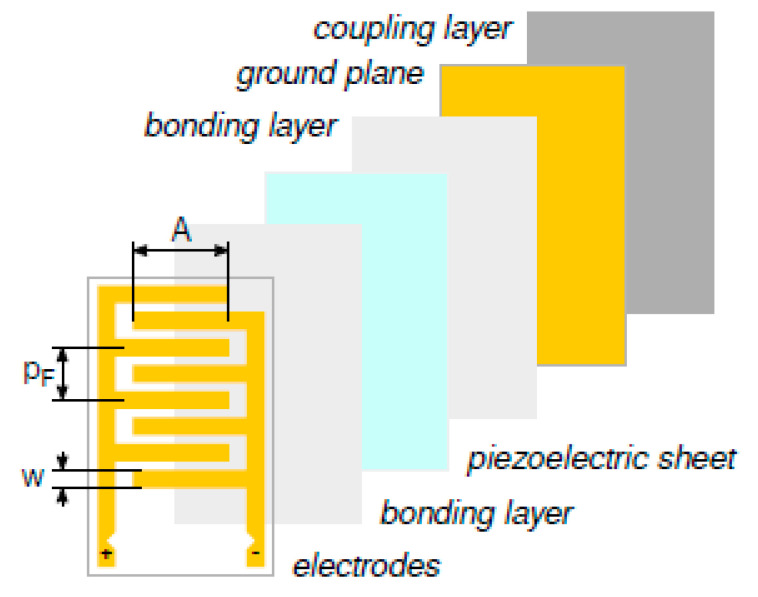
Exploded view of an interdigital transducer assembly. “A” is the length of the electrodes (fingers), “p_F_” is the finger pitch and “w” is the finger width.

**Figure 6 sensors-21-02929-f006:**
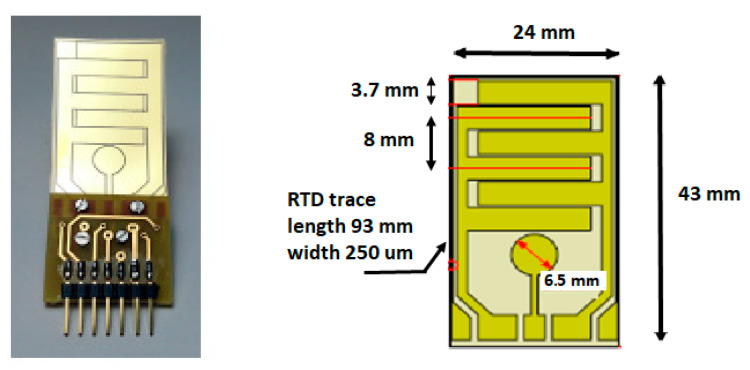
(**Right**) Multifunctional sensor: interdigital transducers (IDT) for active mode with finger pitch 8 mm, circular sensor with diameter 6.5 mm for impact sensing, resistive temperature device (RTD) for temperature monitoring with length (43 + 43 + 24) mm = 110 mm; (**Left**) dimensional drawing of the fabricated device by laser ablation of the metallization.

**Figure 7 sensors-21-02929-f007:**
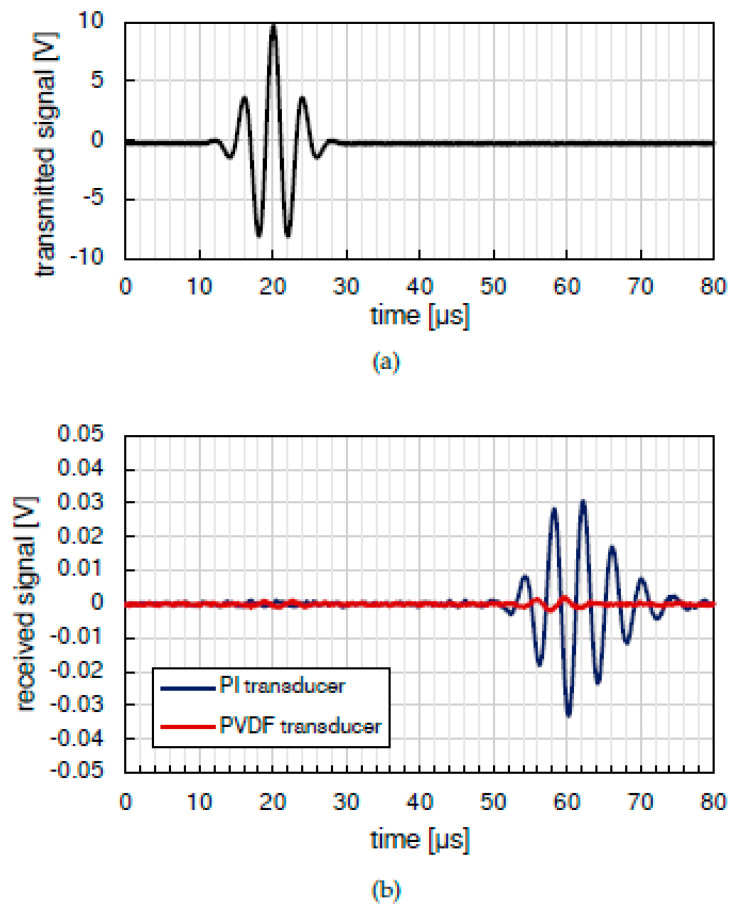
Experimental sensitivity comparison of the circular element with a commercial PZT sensor of same class: (**a**) transmitted Morlet with central frequency 250 kHz; (**b**) signals received from the two sensors (PI blue color and PVDF red color).

**Figure 8 sensors-21-02929-f008:**
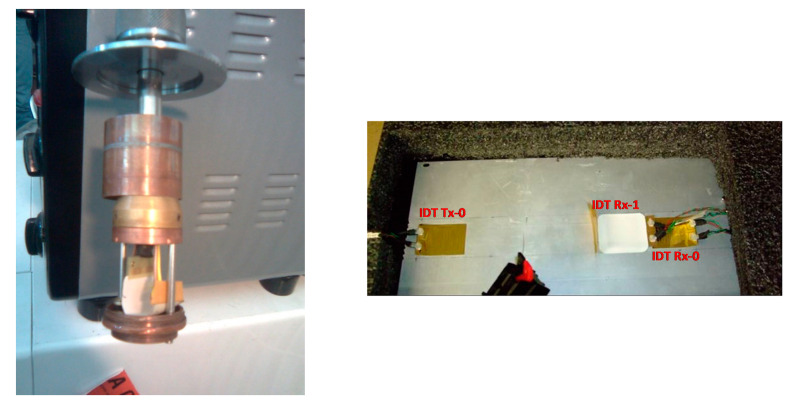
(**Left**) Piezopolymer sensors introduced in the cryogenic chamber. (**Right**) Experimental set up with two piezopolymer transducers in pitch-catch configuration for comparison of performance before and after the cryogenic treatment.

**Figure 9 sensors-21-02929-f009:**
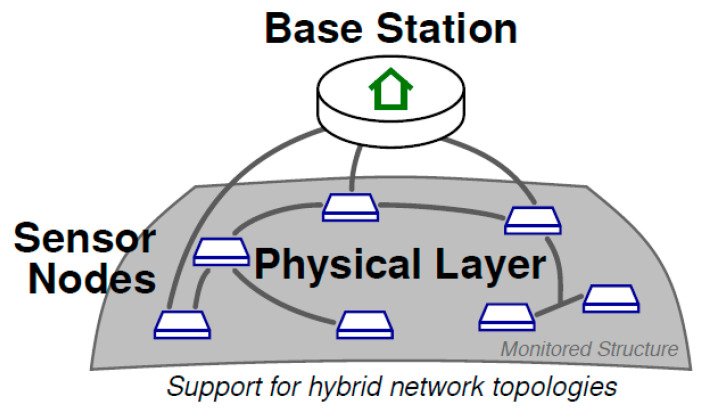
Graphical representation of a wired sensor network for SHM.

**Figure 10 sensors-21-02929-f010:**
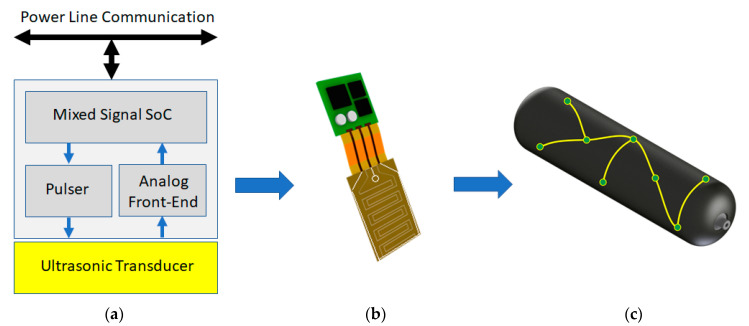
A wired sensor network based on autonomous sensor node design. In the example each node is equipped with ab ultrasonic transducer for active and passive ultrasonic guided waves (UGW) operation: (**a**) node electronic block scheme; (**b**) node rendering; and, (**c**) rendering of a possible application to a Composite Overwrapped Pressure Vessel (COPV) equipped with a wired sensor network.

**Figure 11 sensors-21-02929-f011:**
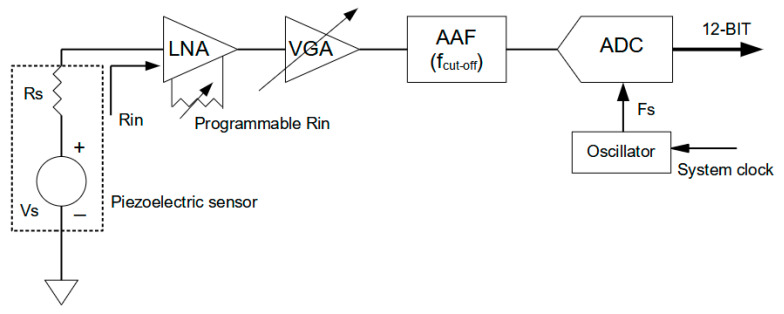
Programmable single channel AFE for signal conditioning of piezoelectric sensor.

**Figure 12 sensors-21-02929-f012:**
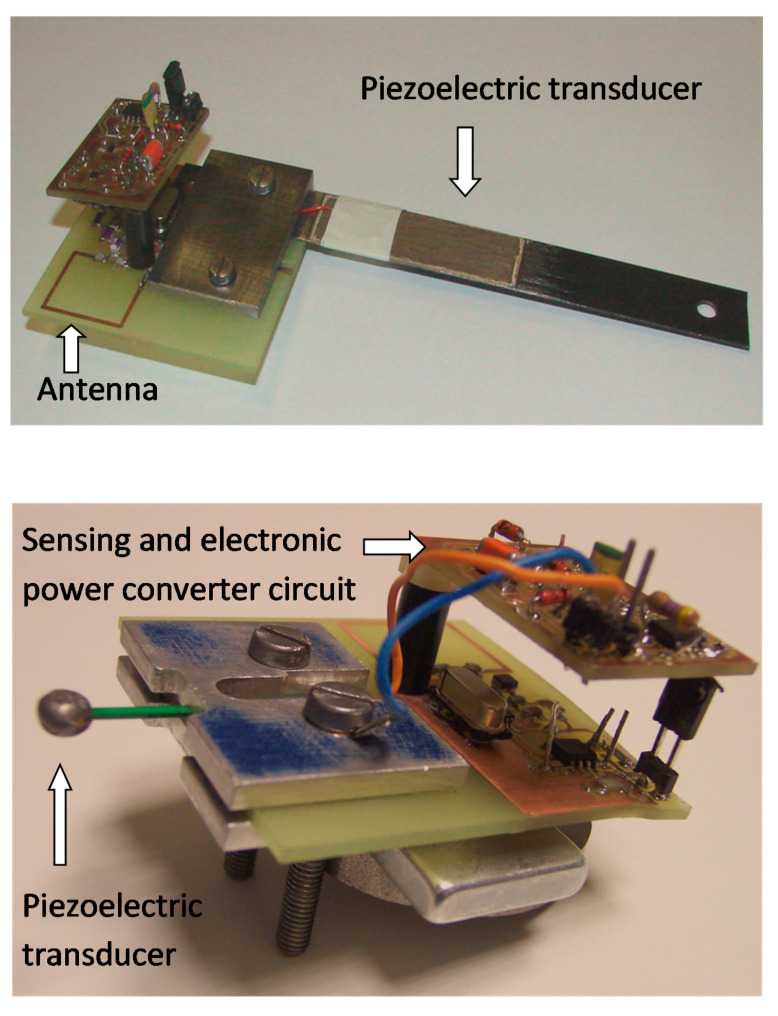
The realized prototype of the autonomous sensor module with a thick-film piezoelectric converter (**top**) and with a commercial piezoelectric converter (**bottom**) (adapted from [141] with authors permission).

**Figure 13 sensors-21-02929-f013:**
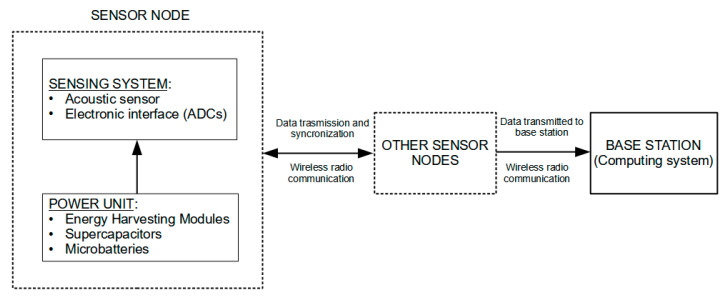
A block diagram of a wireless autonomous sensor network for SHM connected to a base station.

**Table 1 sensors-21-02929-t001:** Characteristics of Single Element Piezoelectric Sensors.

Type	A	B	C
Model	Circular_PVDF	P-876.SP1 DuraAct	SML-SP-1/4-0
Manufacturer	By authors (Precision Acoustics material)	Physik Instrumente	Acellent
Capacitance	86 pF	8 nF +/−20%	1.1 nF
Thickness piezoelectric element [µm]	110	200	140
Material	Piezo-polymer	Piezo-ceramic	Piezo-ceramic
Shape	Circular	Rectangular	Circular
Dimensions [mm]	Diameter 6.5	16 × 13	6
Operation temperature Range	−80 °C, +50 °C	−20 °C, +150 °C	−40 °C, +105 °C
Acoustic Impedance [MRayl]	2.7	30	33

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
