# Peer review of "Ultrasonic Guided-Waves Sensors and Integrated Structural Health Monitoring Systems for Impact Detection and Localization: A Review"

_sensors, 2021, doi:10.3390/s21092929_

Round 1
Reviewer 1 Report
The authors provide a very comprehensive review of current techniques, challenges and possible solutions concerning guided wave testing. Both "passive" and "active" approaches are considered, though the paper focuses more on the former. The considered aspects range from the choice of sensors to the electronics that drive them, though the paper also briefly touches on the signal processing techniques. Overall this work will provide a very useful reference to readers in the guided wave testing fields.
There appears to be some formatting issues with the provided PDF, which also caused a few figures to be cut, so the authors are invited to make sure those issues are solved before the final submission. Also, the authors are encouraged to perform a careful reread of the paper since a fair number of typos and/or grammatical errors can be found. I have noted some of those in the following comments.
1. page 15 line 281: "Zeng et al [29]" it appears that this reference might actually be number [47] in the bibliography
2. page 29 line 556: "description OF this system"
3. page 32 line 602: suggested to rewrite as "following works by Capineri et al 602 [72] and Mamishev et al [73] have developed the fabrication technology, while"
4. page 32 line 605: "simplifies" rather than "make easier"
5. page 38 line 718: "should BE equal or smaller"
6. page 39 line 745: the link to Yun's thesis should be provided in the bibliography rather than in the main text
7. page 40 line 756: "mitigates" rather than "mitigate"
8. page 48 line 930: "started point" should be "start point"
9. same line as above: "taking advantages" should be "taking advantage"
10. page 51 line 973: "structured" should be "structures"
11. page 51 line 990: "made" should be "make"
12. page 54 line 1043: "mainly due to" rather than "due mainly for"
Author Response
Please find the attached file with point-to-point replies.
Thanks again for the extensive revision work.
Reviewer 2 Report
The submitted manuscript presents a broad overview of ultrasonic sensors used guided waves sensing technology for impact damage detection in structures with a focus to the hardware solutions. The manuscript discusses numerous issues related to ultrasonic sensors, their properties and parameters, materials and configurations as well as algorithms of signal conditioning and post-processing. The manuscript contains a lot of interesting and useful information, however, due to the language deficiencies and inappropriate structuring it is difficult to follow. It needs substantial revision according to the comments below and re-review before considering it for a publication.
1) The manuscript needs proof-reading to correct numerous grammar and punctuation errors in the manuscript as well as correcting the style of some sentences. Moreover, “damage” cannot be used as a plural (check the dictionaries), thus the form “damages” is incorrect in the sense it is presented in the manuscript. Please correct in the whole manuscript.
2) The manuscript need to be formatted according the guidelines of the journal, since in present formatting it is difficult to read it.
3) Lines 45-47: “Breakages due to fatigue due to defects, mechanical and thermal stresses, impacts with objects, etc. are all possible causes of damaging.” – I cannot agree with this sentence, there are many other possible damage types not included in this list, please see the literature for this purpose.
4) The statement in line 54: „SHM, unlike Non-Destructive Testing (NDT) …” assumes that NDT is described previously, but it is not. Please briefly introduce the idea behind NDT first.
5) Line 93: it is assumed to use notation “(1)-(9)” to cite formulas, and considering that the formulas in the manuscript are noted in the same way, the notation in line 93 is ambiguous. Same in line 403. Please change the way of citing the list components.
6) Please explain all the abbreviations at their first appearance. There are the cases when the abbreviations are not properly explained in the manuscript, please double-check the manuscript for such cases.
7) Presenting a discussion on the UGW SHM systems it is worth to mention the application of various algorithms used for inspection of impact damage, e.g. MUSIC and RAPID. Some of the studies using these approaches were applied to aircraft structures as well, thus, they fit the discussed applications in aircraft industry.
8) The sections should be appropriately renumbered, since section 1 appears two times: in lines 36 and 159.
9) Since the authors studying the influence of impacts on the structural response, it is essential to extend the description provided in section 1.2 by appropriate overview on damage mechanisms in structures typical for the defined speed ranges, which also have an influence on the resulting wave phenomena.
10) To make it easier to the unexperienced reader to follow the manuscript, it is likely to add schemes of the described phenomena, especially in the section 1 (this one started in line 159).
11) Figures 3, 10-12 requires appropriate formatting.
12) Lines 273-274: CWT probably means continuous wavelet transform, please correct.
13) The description on CWT and STFT is unclear and need further explanations. Please provide, in particular, the explanation why the time-frequency methods need to be considered and provide appropriate examples (widely discussed in the literature) on application of processing algorithms based on the mentioned transforms.
14) Line 335: please explain the meaning of the d35 coefficient.
15) Section 1.4.1 seems to be unnecessarily isolated, since there is no other subsections at this level in section 1.4. It is recommend to incorporate it into section 1.4.
16) The description on advances on signal processing techniques for anisotropic structures need to be significantly extended, since in the present form it just presents a few cases without the necessary state-of-the-art review and analysis of currently applied methods in the matter.
17) The manuscript is focused on UGW, however, at the introductory parts related to SHM it is worth to mention or extend the descriptions on a variety of other approaches used in practice, like the approaches based on the electromechanical impedance and electrical resistance measurements, vibration-based SHM systems (also using the laser vibrometry mentioned in the manuscript) with the developed methods of processing of mode shapes, fiber optic technologies, comparative vacuum monitoring, etc.
18) The provided references to the webpages in lines 410-411, 746, 953, 955 need to be moved to the reference list and numbered in the manuscript body as other references.
19) The description of tests described in section 2.4 need to be enriched with the detailed description of the testing setups.
20) Please double-check the correctness of the names of the cited authors, some of them require correction.
21) The formulas (2)-(6) need to be appropriately formatted according the guidelines, and if the authors do not cite these formulas in the text, their numbering should be omitted.
22) The section 5 is inadequately short with respect to the others, and considering the great variety of studies on the AI and ML in SHM must be extended with more examples and discussion.
Author Response

(The authors gave the same response as above.)

Reviewer 3 Report
The submitted review article “sensors-1160577-v1” entitled: “Ultrasonic Guided-Waves Sensors and integrated Structural Health Monitoring systems for impact detection and localization: A review” is an effort to report the recent research developments on the integration of sensors and Structural Health Monitoring (SHM) systems with the electronic interface for impact monitoring and detection Several types and materials of sensors implemented on host structure are presented. This version of the paper seems weak for publication in Sensors Journal. The following comments and suggestions are raised for authors’ reference:
- The specific tasks and the objectives of this study are fairly defined, and, therefore more convincing motivations of this review report are rather required.
- Manuscript is rather long and could be shortened by providing Tables that summarize and compare the characteristics, the parameters and the differences between the examined sensors and SHM systems. Several discussions seem rather shallow lacking coherence and failing to present a comprehensive review report and an in-depth commentary.
- Some Figures are briefly discussed (for example Fig. 2, 6, 7, 11) and some other Figures seem inconsistent, or fairly relative to the review report. Further, several Figures are trimmed (outside page margins).
- A critical review of the reported research topics would enhance the research significance and contribution of this paper.
Author Response

(The authors gave the same response as above.)

Reviewer 4 Report
The paper is a review of the Ultrasonic Guided-Waves Sensors and the integrated Structural Health Monitoring system for impact detection and localization. The discussion is based on the technological development which this SHM methods have had in recent years. The paper structure is focussed on a state of art analysis that aim to discuss the main aspect of the UGW sensors for impact detection and localization describing in brief and exhaustive way many references found in scientific literature. After introducing the physical phenomena and the quantities, which play a big role in impact phenomena, the discussion moves on deep analysis of the hardware and software components. The main aspect of the paper is the description of the several sensor technologies currently present in the literature and what advantages and disadvantages they could bring, in relation to the different phenomena investigated. In this context the authors, try to provide a careful analysis on the front-end electronics, data transmission typology both in term of wired and wireless technology and in term of online and offline signal processing. The content is then enriched by introducing some considerations on the development of recent methodologies based on artificial intelligence and machine learning with the aim of identify damage in the investigated structures subjected to impact excitations. The paper is overall well written and it is supported by an extensive bibliography, however before publication is mandatory that it undergo to minor review in order to solve some serious problems, especially in terms of text formatting, which are listed below:
- Figure 3-10-11-12 need to be replaced with figure of appropriate size as they do not fit into the textual box and are cut laterally with consequent loss of information;
- At line 504 there is a “@” in the text. What does it mean?
- Some acronyms as COTS (line 739), CMRR (line 757) and CFRP (line 342) are not defined both in the text and in the appendix;
- Some acronyms as ANN (line 292), CMUT (line 785), PMUT (line 785) are defined only in appendix. It would be worth it to define the meaning before the acronym utilizations;
- The acronym DToA at line 637 is written as DTOA, where O is capitalized;
- Some acronyms in text are defined capitalizing the first letter other not as in line 274; please be consistent.
Author Response

(The authors gave the same response as above.)

Reviewer 5 Report
Authors proposed the review of the structure health monitoring systems of the piezoelectric sensors and discussed the wire/wireless sensor network. Authors included the concepts of too many sensor and system integrations together. Therefore, it is a kind of being confused to understand whole stories in the abstract and conclusion sections. Therefore, authors need to be concise in the abstract section. For my personal opinion, contents need to be checked more carefully. Authors also need to add some missing references. In addition, there are some grammar mistakes so it is better to ask your colleague faculty to check the English grammar or use professional English services in entire manuscript because some units and values need to have some spaces and countless words cannot use "the". Therefore, the manuscript needs to be revised.
1. In all the sections, please reduce space between each sentence. Please do not use double spacing.
2. Please do not use all capital letters for journal in the references.
3. Please use abbreviated journal names in the reference sections.
4. Authors need to provide abbreviated conference name in reference.
3. Authors need to provide the city and country and date information in Ref. [14].
5. Please provide the author contribution, data availability, and acknowledgement sections.
6. Figures 1 and 3 labels are not clear to be seen.
7. Please increase section size of 1.3 and 1.4. Please check others.
8. Please do not use color in Table 1.
9. Please use proper minus sign for all digits. Please check others.
10. Please use the reference in Lines 410 and 411.
11. Figure 6 (b) has low quality.
12. Please use thick letter for Figure 9 label.
13. Please add the reference (velocity vi can be calculated by knowing the kinetic energy Ek and the mass m of the impacting object) with the reference (Iben Jr, Icko, and Alexander V. Tutukov. "Supernovae of type I as end products of the evolution of binaries with components of moderate initial mass (M not greater than about 9 solar masses)." The Astrophysical Journal Supplement Series 54 (1984): 335-372.) or anther reference.
14. Please add the reference (the region of interest (ROI) manually scanned of by a trained operator; main differences are found for the signal processing adopted both for passive and active mode operation of the SHM system.) with the reference ( Chandrasekaran, Srinivasan. Structural health monitoring with application to offshore structures. World Scientific, 2019.) or anther reference.
15. Please add the reference (The accuracy of the impact point estimation depends on the estimates of the guided modes velocity and the measured differential time of arrival (DToA) among the sensors) with the reference (Qing, Xinlin, et al. "Piezoelectric transducer-based structural health monitoring for aircraft applications." Sensors 19.3 (2019): 545. ) or anther reference.
16. Please add the reference (Piezoelectric sensors are common devices for the passive detection of impacts on the structure.) with the reference (Giurgiutiu, Victor, and Giola Santoni-Bottai. "Structural health monitoring of composite structures with piezoelectric-wafer active sensors." AIAA journal 49.3 (2011): 565-581. ) or anther reference.
17. Please add the reference (In Figure 9 are shown the main electronic components of a programmable single channel AFE and we include a numerical example for the evaluation of performance) with the reference (You, K.; Choi, H. Wide Bandwidth Class-S Power Amplifiers for Ultrasonic Devices. Sensors 2020, 20, 290.)
18. Please check Figure 3 because of missing contents.
19. In Line 368, there is missing word after Dynamic.
20. In Table 1, please use spacing between the digit and unit such as 86pF.
21. Two Figures 12 are too small so it is better put those Figures in the top and bottom positions.
22. Please use dash and minus symbols properly.
Author Response

(The authors gave the same response as above.)

Round 2
Reviewer 2 Report
The authors revised the manuscript according the most of the comments given in the first round review report and significantly improved it, however, some comments and questions need to be reconsidered and revised.
1) The language of the manuscript still needs improvements due to the presence of numerous grammar errors. The proof-reading of the manuscript is recommended.
2) The formatting of the manuscript is still not appropriate, please check the templates.
3) Both MUSIC and RAPID algorithms (as well as some other, as the authors mentioned in their response) are appropriate also for a localization. I realize that the manuscript is already quite large and adding new content will extend it even more. However, if the authors discuss the topic of UGW in SHM context, it is important to provide a comprehensive overview, especially taking into consideration that it is a review paper. The algorithms used for damage detection and localization are directly correlated with the hardware systems, since the type and configuration of sensors has an influence on the performance of these algorithms.
4) I understand the decision of the authors not to extend the manuscript in terms of discussion the advances of signal processing techniques (comment 16 in the previous review report), however, again, because it is the review paper, it needs to be at least linked with the appropriate sources, where a reader may find necessary information. The authors may consider referring the reader to several papers important in this field or books that contain a comprehensive analysis in this matter.
Reviewer 3 Report
The revised review article “sensors-1160577-v2” entitled: “Ultrasonic Guided-Waves Sensors and integrated Structural Health Monitoring systems for impact detection and localization: A review” has been improved. However, some revisions and amendments are still required before publication. Based on the comments of the first review, the following are reported:
Comment 1: It has been considered successfully: The specific tasks and the objectives of this study are well defined, and convincing motivations of this review report have been added.
Comment 2. Needs revision: The advances in sensor technology with low-power mixed-signal electronics are fairly reported and discussed. An in-depth commentary is rather required.
Comment 3. It has been considered successfully: Figures are adequately discussed and improved.
Comment 4. Needs revision: A critical review of the reported research topics that would enhance the research significance and contribution of this paper is missing. A synopsis Table would help in this direction.
Reviewer 5 Report
Authors defended all the issues very carefully so the manuscript is accepted.
